# Task-Aware Information Routing from Common Representation Space in Lifelong Learning

**Prashant Bhat[1], Bahram Zonooz[1,2]\* & Elahe Arani[1,2]\***
[1]Advanced Research Lab, NavInfo Europe, Netherlands
[2]Dep. of Mathematics and Computer Science, Eindhoven University of Technology, Netherlands
`prashant.bhat@navinfo.eu`, `{bahram.zonooz, e.arani}@gmail.com`

## Abstract

Intelligent systems deployed in the real world suffer from catastrophic forgetting when exposed to a sequence of tasks. Humans, on the other hand, acquire, consolidate, and transfer knowledge between tasks that rarely interfere with the consolidated knowledge. Accompanied by self-regulated neurogenesis, continual learning in the brain is governed by a rich set of neurophysiological processes that harbor different types of knowledge, which are then integrated by conscious processing. Thus, inspired by the Global Workspace Theory of conscious information access in the brain, we propose TAMiL, a continual learning method that entails task-attention modules to capture task-specific information from the common representation space. We employ simple, undercomplete autoencoders to create a communication bottleneck between the common representation space and the global workspace, allowing only the task-relevant information to the global workspace, thus greatly reducing task interference. Experimental results show that our method outperforms state-of-the-art rehearsal-based and dynamic sparse approaches and bridges the gap between fixed capacity and parameter isolation approaches while being scalable. We also show that our method effectively mitigates catastrophic forgetting while being well-calibrated with reduced task-recency bias[1].

## 1 Introduction

Deep neural networks (DNNs) deployed in the real world are normally required to learn multiple tasks sequentially and are exposed to non-stationary data distributions. Throughout their lifespan, such systems must acquire new skills without compromising previously learned knowledge. However, continual learning (CL) over multiple tasks violates the i.i.d. (independent and identically distributed) assumption on the underlying data, leading to overfitting on the current task and catastrophic forgetting of previous tasks. The menace of catastrophic forgetting occurs due to the stability-plasticity dilemma: the extent to which the system must be stable to retain consolidated knowledge and be plastic to assimilate new information (Mermillod et al., 2013). As a consequence of catastrophic forgetting, performance on previous tasks often drops significantly; in the worst case, previously learned information is completely overwritten by the new one (Parisi et al., 2019).

Humans, however, excel at CL by incrementally acquiring, consolidating, and transferring knowledge across tasks (Bremner et al., 2012). Although there is gracious forgetting in humans, learning new information rarely causes catastrophic forgetting of consolidated knowledge (French, 1999). CL in the brain is governed by a rich set of neurophysiological processes that harbor different types of knowledge, and conscious processing integrates them coherently (Goyal & Bengio, 2020). Self-regulated neurogenesis in the brain increases the knowledge bases in which information related to a task is stored without catastrophic forgetting (Kudithipudi et al., 2022). The global workspace theory (GWT) (Baars, 1994; 2005; Baars et al., 2021) posits that one of such knowledge bases is a

---

\*Shared last author

[1]Code is available at: `https://github.com/NeurAI-Lab/TAMiL`

common representation space of fixed capacity from which information is selected, maintained, and shared with the rest of the brain. When addressing the current task, the attention mechanism creates a communication bottleneck between the common representation space and the global workspace and admits only relevant information in the global workspace (Goyal & Bengio, 2020). Such a system enables efficient CL in humans with systematic generalization across tasks (Bengio, 2017).

Several approaches have been proposed in the literature that mimic one or more neurophysiological processes in the brain to address catastrophic forgetting in DNN. Experience rehearsal (Ratcliff, 1990) is one of the most prominent approaches that mimics the association of past and present experiences in the brain. However, the performance of rehearsal-based approaches is poor under low buffer regimes, as it is commensurate with the buffer size (Bhat et al., 2022a). On the other hand, parameter isolation methods (Rusu et al., 2016) present an extreme case of neurogenesis in which a new subnetwork is initialized for each task, thus greatly reducing task interference. Nevertheless, these approaches exhibit poor reusability of parameters and are not scalable due to the addition of a large number of parameters per task. Therefore, the right combination of the aforementioned mechanisms governed by GWT could unlock effective CL in DNNs while simultaneously encouraging reusability and mitigating catastrophic forgetting.

Therefore, we propose *Task-specific Attention Modules in Lifelong learning (TAMiL)*, a novel CL approach that encompasses both experience rehearsal and self-regulated scalable neurogenesis. Specifically, TAMiL learns by using current task samples and a memory buffer that represents data from all previously seen tasks. Additionally, each task entails a task-specific attention module (TAM) to capture task-relevant information in CL, similar to self-regulated neurogenesis in the brain. Reminiscent of the conscious information access proposed in GWT, each TAM acts as a bottleneck when transmitting information from the common representation space to the global workspace, thus reducing task interference. Unlike self-attention in Vision Transformers, we propose using a simple, undercomplete autoencoder as a TAM, thereby rendering the TAMiL scalable even under longer task sequences. Our contributions are as follows:

- We propose TAMiL, a novel CL approach that entails both experience rehearsal and self-regulated scalable neurogenesis to further mitigate catastrophic forgetting in CL.
- Inspired by GWT of conscious information access in the brain, we propose TAMs to capture task-specific information from the common representation space, thus greatly reducing task interference in Class- and Task-Incremental Learning scenarios.
- We also show a significant effect of task attention on other rehearsal-based approaches (e.g. ER, FDR, DER++). The generalizability of the effectiveness of TAMs across algorithms reinforces the applicability of GWT in computational models in CL.
- We also show that TAMiL is scalable and well-calibrated with reduced task-recency bias.

## 2 RELATED WORKS

**Rehearsal-based Approaches:** Continual learning over a sequence of tasks has been a long-standing challenge, since learning a new task causes large weight changes in the DNNs, resulting in overfitting on the current task and catastrophic forgetting of older tasks (Parisi et al., 2019). Similar to experience rehearsal in the brain, early works attempted to address catastrophic forgetting through Experience-Replay (ER; Ratcliff (1990); Robins (1995)) by explicitly storing and replaying previous task samples alongside current task samples. Function Distance Regularization (FDR; Benjamin et al. (2018)), Dark Experience Replay (DER++; Buzzega et al. (2020)) and CLS-ER (Arani et al., 2022) leverage soft targets in addition to ground truth labels to enforce consistency regularization across previous and current model predictions. In addition to rehearsal, DRI (Wang et al., 2022) utilizes a generative model to augment rehearsal under low buffer regimes. On the other hand, $Co^2L$ (Cha et al., 2021), TARC (Bhat et al., 2022b) and ER-ACE (Caccia et al., 2021a) modify the learning objective to prevent representation drift when encountered with new classes. Given sufficient memory, replay-based approaches mimic the association of past and present experiences in humans and are fairly successful in challenging CL scenarios. However, in scenarios where buffer size is limited, they suffer from overfitting (Bhat et al., 2022a), exacerbated representation drift (Caccia et al., 2021b) and prior information loss (Zhang et al., 2020) resulting in aggravated forgetting of previous tasks.

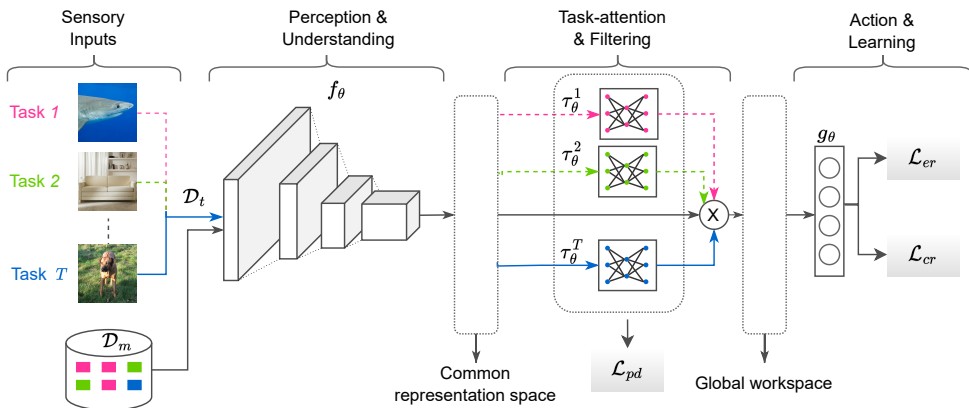

Figure 1: The proposed method, TAMiL, incorporates both experience rehearsal and self-regulated scalable neurogenesis. Firstly, the current task data, $\mathcal{D}_t$, is mapped to a common representation space using $f_\theta$. The corresponding Task-Specific Attention Module (TAM) then captures task-specific information contained in $\mathcal{D}_t$ and applies transformation coefficients to select features important for the current task, thereby preventing interference between tasks. A matching criterion is used as an ignition event to trigger a particular TAM for both buffered and test samples.

**Evolving Architectures:** In addition to experience rehearsal, CL in the brain is mediated by self-regulated neurogenesis that scale up the number of new memories that can be encoded and stored without catastrophic forgetting (Kudithipudi et al., 2022). Similarly in DNNs, Progressive Neural Networks (PNNs; Rusu et al. (2016)) instantiate a new subnetwork for each task with lateral connections to previously learned frozen models. Several works have been proposed to address the issue of scalability in PNNs: CCLL (Singh et al., 2020) employed a fixed capacity model and reused the features captured on the first task by performing spatial and channel-wise calibration for all subsequent tasks. DEN (Yoon et al., 2018) proposed a dynamically expandable network using selective retraining, network expansion with group sparsity regularization, and neuron duplication. Similarly, CPG (Hung et al., 2019a) proposed an iterative approach with pruning of previous task weights followed by gradual network expansion while reusing critical weights from previous tasks. MNTDP (Veniat et al., 2020) employed a modular learning approach to transfer knowledge between related tasks while sublinearly scaling with the number of tasks. Although these approaches grow drastically slower than PNNs, they require task identity at inference time. On the other hand, Mendez & Eaton (2020) explicitly captured compositional structures in lifelong learning thereby enabling their reusability across tasks. However, this requires joint training of subset of tasks to learn initial generalizable compositional structures. Requiring task-identity at inference and joint training of subset of tasks at initialization are an impediment for deploying these CL systems in the real world.

By contrast, several other methods (e.g. NISPA (Gurbuz & Dovrolis, 2022), CLNP (Golkar et al., 2019), PackNet (Mallya & Lazebnik, 2018), PAE (Hung et al., 2019b)) proposed dynamic sparse networks based on neuronal model sparsification with fixed model capacity. Similar to the brain, these models simultaneously learn both connection strengths and a sparse architecture for each task, thereby isolating the task-specific parameters. However, these methods suffer from capacity saturation in longer task sequences, thus rendering them inapplicable to real-world scenarios.

We propose TAMiL, a CL method that, while drawing inspiration from GWT, successfully combines rehearsal and parameter isolation with little memory overhead (Table 4), superior performance (Table 1) without capacity saturation, and without requiring task identity at inference time. To the best of our knowledge, our work is the first to study the GWT-inspired CL approach to effectively mitigate catastrophic forgetting.

## 3 PROPOSED METHOD

We consider a CL setting in which multiple sequential tasks $t \in \{1, 2, .., T\}$ are learned by the model $\Phi_\theta$ one at a time. Each task is specified by a task-specific distribution $\mathcal{D}_t$ with $\{(x_i, y_i)\}_{i=1}^N$ pairs.

In this training paradigm, any two task-specific distributions are disjoint. The model $\Phi_\theta$ consists of a backbone network $f_\theta$ and a classifier $g_\theta$ that represents classes belonging to all tasks. The learning objective in such an CL setting is to restrict the empirical risk of all tasks seen so far:

$$\mathcal{L}_t = \sum_{t=1}^{T_c} \mathbb{E}_{(x_i,y_i)\sim\mathcal{D}_t} \left[ \mathcal{L}_{ce}(\sigma(\Phi_\theta(x_i)), y_i) \right], \tag{1}$$

where $\mathcal{L}_{ce}$ is a cross-entropy loss, $t$ is the current task, and $\sigma$ is the softmax function. Critically, sequential learning causes significant weight changes in $\Phi_\theta$ in subsequent tasks, resulting in catastrophic forgetting of previous tasks and overfitting on the current task if $\Phi_\theta$ is trained on each task only once in its lifetime without revisiting them. To mitigate catastrophic forgetting in CL, we employ experience rehearsal along with consistency regularization through episodic replay. Central to our method are the Task-specific Attention Modules (TAMs) that attend to important features of the input. We define two representation spaces, namely common representation space and global workspace that are spanned by mapping functions $\mathcal{M}_f : \mathbb{R}^{B,H,W,C} \to \mathbb{R}^D$ and $\mathcal{M}_{TAM} : \mathbb{R}^D \to \mathbb{R}^D$ where $D$ denotes the dimension of the output Euclidean space. $\mathcal{M}_f$ is a set of possible functions that the encoder $f_\theta$ can learn, while $\mathcal{M}_{TAM}$ denotes a set of functions represented by TAMs. We use simple undercomplete autoencoders as task-specific attention modules that can act as feature selectors. We describe each of these components shown in Figure 1 in the following sections.

## 3.1 Episodic replay

To preserve knowledge of previous tasks, we seek to approximate previous data distributions $\mathcal{D}_{t\in\{t:1\leq i<T_c\}}$ through a memory buffer $\mathcal{D}_m$ with reservoir sampling (Vitter, 1985). Each sample in $\mathcal{D}_t$ has the same probability of being represented in the buffer and replacements are performed randomly. At each iteration, we randomly sample from $\mathcal{D}_m$ and replay them along with $\mathcal{D}_t$. Therefore, the objective function in Eq. 1 can be conveniently modified as follows:

$$\mathcal{L}_{er} = \mathcal{L}_{T_c} + \alpha \mathbb{E}_{(x_j,y_j)\sim\mathcal{D}_m} \left[ \mathcal{L}_{ce}(\sigma(\Phi_\theta(x_j)), y_j) \right], \tag{2}$$

where $\alpha$ is a balancing parameter. Experience rehearsal improves stability that is commensurate with the ability of $\mathcal{D}_m$ to approximate past distributions. In scenarios where buffer size is limited, the CL model learns sample-specific features rather than capturing class- or task-wise representative features, resulting in poor performance under low buffer regimes. As soft targets carry more information per training sample than hard targets, we therefore employ consistency regularization Bhat et al. (2022a) (Action & Learning in Figure 1) to better preserve the information from previous tasks. We straightforwardly define consistency regularization using mean squared error as follows:

$$\mathcal{L}_{cr} \triangleq \mathbb{E}_{(x_j,y_j,z_j)\sim D_m} \|z_j - \Phi_\theta(x_j)\|_2^2 \tag{3}$$

where $z_j$ represents the pre-softmax responses of an Exponential Moving Average (EMA) of the CL model. Alternatively, $z_j$ from previous iterations can also be stored in the buffer.

## 3.2 Task-specific Attention Modules In Lifelong learning (TAMiL)

Reminiscent of the conscious information access proposed in GWT, we propose task-specific attention modules (TAMs) (Task attention & Filtering in Figure 1) to capture task-relevant information in CL[2]. Following Stephenson et al. (2020), we believe that the common representation space spanned by $\mathcal{M}_f$ captures the relevant generic information for all tasks, while the TAMs capture task-specific information. The choice of these attention modules should be such that there is enough flexibility for them to capture task-relevant information, and they are diverse enough to differentiate between tasks during inference while still rendering the CL model scalable in longer task sequences. To this end, we propose using simple undercomplete autoencoders as TAMs. Each of these TAMs consists of two parts $\tau_\theta^i = \{\tau_\theta^{ie}, \tau_\theta^{is}\}$, where $\tau_\theta^{ie}$ acts as a feature extractor and $\tau_\theta^{is}$ as a feature selector. The feature extractor learns a low-dimensional subspace using a linear layer followed by ReLU activation. On the other hand, the feature selector learns task-specific attention using another linear layer followed by sigmoid activation. The bottleneck in the proposed TAMs achieves twin objectives: (i)

---

[2]TAMs have similar structure as autoencoders but do not reconstruct the input

it inhibits TAMs from reconstructing their own input, while (ii) it reduces the number of parameters required to learn task-relevant information. Similar to neurogenesis in the brain, TAMs encode and store task-specific attention while still being scalable to a large number of tasks.

To effectively leverage the functional space of TAMs, we seek to maximize pairwise discrepancy loss between output representations of the TAMs trained so far:

$$\mathcal{L}_{pd} \triangleq \sum_{t=1}^{T_c-1} \mathbb{E}_{x \sim D_t} \|\sigma(\tau_\theta^{T_c}(r)) - stopgrad(\sigma(\tau_\theta^t(r)))\|_{p=1} \tag{4}$$

where $r = f_\theta(x)$ is the representation in the common representation space. As a stricter pairwise discrepancy could result in capacity saturation and reduce flexibility to learn new tasks, we employ the softmax function $\sigma(.)$ while enforcing the diversity between TAMs. We also update the gradients of only the current TAM $\tau_\theta^t$ to avoid overwriting the previous task attention using $stopgrad(.)$. Without Eq. 4, multiple TAMs can be very similar, reducing their effectiveness as task-specific attention.

### 3.3 PUTTING IT ALL TOGETHER

TAMiL consists of a CL model $\Phi_\theta = \{f_\theta, \tau_\theta, g_\theta\}$ where $f_\theta$ represents a feature extractor (e.g. ResNet-18), $\tau_\theta = \{\tau_\theta^k \mid k \leq t\}$ is a set of TAMs up to the current task $t$, and the classifier $g_\theta$ represents classes belonging to all tasks. Analogous to the common representation space proposed in GWT, we employ $f_\theta$ as a common representation space to capture sensory information $\mathcal{D}_t$ from all tasks sequentially. For each task, a new TAM is initialized that acts as a feature selector by attending to features important for the given task. The intuition behind placing TAMs higher up in the layer hierarchy is as follows: the early layers of DNNs capture generic information, while the later layers memorize due to the diminishing dimension and radius of the manifold (Stephenson et al., 2020; Baldock et al., 2021). Therefore, redundancy in the later layers is desirable to reduce catastrophic forgetting while maximizing reusability.

The goal of TAMs is to act as a task-specific bottleneck through which only task-relevant information is sent through the global workspace spanned by $\mathcal{M}_{TAM}$. Specifically, during CL training, the corresponding TAM learns to weigh the incoming features according to the task identifier using the current task data $\mathcal{D}_t$. The output of the corresponding TAM termed transformation coefficients are then applied to the features of the common representation space using element-wise multiplication. Furthermore, we enforce the pairwise discrepancy loss in Eq. 4 to ensure the diversity among TAMs. On the downside, since each TAM is associated with a specific task, inferring a wrong TAM for the test samples can result in sub-par performance on the test set.

In the brain, information is not always processed consciously unless there is sufficient activation in the prefrontal region, resulting in an ignition event (Juliani et al., 2022). Analogously, we emulate the ignition event with a matching criterion using buffered samples from $\mathcal{D}_m$. During training, for each buffered sample, we infer the identity of the task by computing the mean squared error between the feature $r_m$ of the common representation space and the output of each of the TAMs seen so far. We select the TAM with the lowest matching criterion as follows:

$$\underset{k \in 1,..,t}{\operatorname{argmin}} \|\tau_\theta^k(r_m) - r_m\|_2^2 \tag{5}$$

where $r_m = f_\theta(x_j), x_j \in \mathcal{D}_m$. Once the right TAM is selected, we apply cross-entropy loss (Eq. 2) and consistency regularization (Eq. 3) on the buffered samples. As the CL model is now trained to select the appropriate TAM, we also use the same criterion during the inference stage. We selected the matching criterion as an ignition event because of its simplicity and lack of additional trainable parameters. However, complex alternatives, such as learning a policy using reinforcement learning, a gating mechanism using Gumbel-softmax, and prototype matching, can also be explored.

In addition to $\mathcal{L}_{pd}$ (Eq. 4), we do not use any other objective on the TAMs to constrain their learning. The final learning objective for the entire CL model is as follows:

$$\mathcal{L} \triangleq \mathcal{L}_{er} + \beta \, \mathcal{L}_{cr} - \lambda \, \mathcal{L}_{pd} \tag{6}$$

Our proposed approach is illustrated in Figure 1 and is detailed in Algorithm 1.

---

**Algorithm 1** Proposed Method

---

**input:** Data streams $\mathcal{D}_t$, Model $\Phi_\theta = \{f_\theta, \tau_\theta, g_\theta\}$, Balancing factors $\alpha$, $\beta$ and $\lambda$
$\quad\quad$ Memory buffer $\mathcal{D}_m \leftarrow \{\}$, TAMs $\tau_\theta \leftarrow \{\}$

1: **for all** tasks $t \in \{1, 2, .., T\}$ **do**
2: $\quad$ $\tau_\theta = \tau_\theta \cup \{\tau_\theta^t\}$
3: $\quad$ **for** minibatch $\{x_i, y_i\}_{i=1}^B \in \mathcal{D}_t$ **do**
4: $\quad\quad$ $\hat{y}_i = g_\theta(\tau_\theta^t(f_\theta(x_i)) \otimes f_\theta(x_i))$
5: $\quad\quad$ Compute $\mathcal{L}_{er} = \frac{1}{B} \sum_B \mathcal{L}_{ce}(\hat{y}_i, y_i)$ $\quad\quad\quad\quad\quad\quad\quad\quad\quad\quad\quad$ ▷ (Equation 2)
6: $\quad\quad$ Compute $\mathcal{L}_{pd}$ $\quad\quad\quad\quad\quad\quad\quad\quad\quad\quad\quad\quad\quad\quad\quad\quad\quad$ ▷ (Equation 4)
7: $\quad\quad$ **if** $\mathcal{D}_m \neq \emptyset$ **then**
8: $\quad\quad\quad$ **for** minibatch $\{x_j, y_j, z_j\}_{j=1}^B \in \mathcal{D}_m$ **do**
9: $\quad\quad\quad\quad$ $r_m = f_\theta(x_j)$
10: $\quad\quad\quad\quad$ $k = \underset{k \in 1,..,t}{\text{argmin}} \|\tau_\theta^k(r_m) - r_m\|_2^2$ $\quad\quad\quad\quad\quad\quad\quad\quad$ ▷ (Equation 5)
11: $\quad\quad\quad\quad$ $\hat{y}_j = g_\theta(\tau_\theta^k(r_m) \otimes r_m)$
12: $\quad\quad\quad\quad$ Compute $\mathcal{L}_{er}$ $+= \frac{\alpha}{B} \sum_B \mathcal{L}_{ce}(\hat{y}_j, y_j)$ $\quad\quad\quad\quad\quad\quad$ ▷ (Equation. 2)
13: $\quad\quad\quad\quad$ Compute $\mathcal{L}_{cr}$ $\quad\quad\quad\quad\quad\quad\quad\quad\quad\quad\quad\quad\quad\quad\quad$ ▷ (Equation 3)
14: $\quad\quad$ Compute $\mathcal{L} \triangleq \mathcal{L}_{er} + \beta \mathcal{L}_{cr} - \lambda \mathcal{L}_{pd}$ $\quad\quad\quad\quad\quad\quad\quad$ ▷ (Equation. 6)
15: $\quad\quad$ Compute the gradients $\frac{\delta\mathcal{L}}{\delta\theta}$ and update the model $\Phi_\theta$
16: $\quad\quad$ Update the memory buffer $\mathcal{D}_m$ $\quad\quad\quad\quad\quad\quad\quad\quad\quad\quad\quad$ ▷ (Algorithm 2)
17: **return** model $\Phi_\theta$

---

## 4 EXPERIMENTAL SETUP

We build on top of the Mammoth (Buzzega et al., 2020) CL repository in PyTorch. We consider two CL scenarios, namely, Class-Incremental Learning (Class-IL) and Task-Incremental Learning (Task-IL). In a Class-IL setting, the CL model encounters mutually exclusive sets of classes in each task and must learn to distinguish all classes encountered thus far by inferring the task identity. On the contrary, the task identity is always provided during both training and inference in the Task-IL scenario. More information on the details of the implementation can be found in Appendix D. In the empirical results, we compare with several state-of-the-art rehearsal-based methods and report average accuracy after learning all the tasks. As our method employs consistency regularization, we also compare it with the popular regularization-based method LwF (Li & Hoiem, 2017). In addition, we provide a lower bound SGD, without any help to mitigate catastrophic forgetting, and an upper bound Joint, where training is done using entire dataset. In the *Oracle* version, for any test sample $x \in \mathcal{D}_t$, we use the task identity at the inference time to select the right TAM.

## 5 RESULTS

Table 1 presents the evaluation of different CL models on multiple sequential datasets. We can make several observations: (i) Across all datasets, TAMiL outperforms all the rehearsal-based baselines considered in this work. As is the case in GWT, TAMs capture task-specific features and reduce interference, thereby enabling efficient CL with systematic generalization across tasks. For example, in the case of Seq-TinyImageNet with buffer size 500, the absolute improvement over the closest baseline is $\sim 10\%$ in both CL scenarios. (ii) The performance improvement in Class-IL is even more pronounced when we know the identity of the task (Oracle version). Notwithstanding their size, this is a testament to the ability of TAMs to admit only relevant information from the common representation space to the global workspace when warranted by a task-specific input. (iii) Given the bottleneck nature of TAMs, the additional parameters introduced in each task are negligible in size compared to the parameter growth in PNNs. However, TAMiL bridges the performance gap between rehearsal-based and parameter-isolation methods without actually incurring a large computational overhead. Given sufficient buffer size, our method outperforms PNNs (e.g. in the case of Seq-CIFAR100 with buffer size 500, our method outperforms the PNNs).

Table 1: Comparison of CL models across various CL scenarios. We provide the average Top-1 (%) accuracy of all tasks after CL training. Forgetting analysis can be found in Appendix C.2.

| Buffer size | Methods | Seq-CIFAR10 | | Seq-CIFAR100 | | Seq-TinyImageNet | |
|---|---|---|---|---|---|---|---|
| | | Class-IL | Task-IL | Class-IL | Task-IL | Class-IL | Task-IL |
| - | SGD | $19.62_{\pm0.05}$ | $61.02_{\pm3.33}$ | $17.49_{\pm0.28}$ | $40.46_{\pm0.99}$ | $07.92_{\pm0.26}$ | $18.31_{\pm0.68}$ |
| | Joint | $92.20_{\pm0.15}$ | $98.31_{\pm0.12}$ | $70.56_{\pm0.28}$ | $86.19_{\pm0.43}$ | $59.99_{\pm0.19}$ | $82.04_{\pm0.10}$ |
| - | LwF | $19.61_{\pm0.05}$ | $63.29_{\pm2.35}$ | $18.47_{\pm0.14}$ | $26.45_{\pm0.22}$ | $8.46_{\pm0.22}$ | $15.85_{\pm0.58}$ |
| | PNNs | - | $95.13_{\pm0.72}$ | - | $74.01_{\pm1.11}$ | - | $67.84_{\pm0.29}$ |
| 200 | ER | $44.79_{\pm1.86}$ | $91.19_{\pm0.94}$ | $21.40_{\pm0.22}$ | $61.36_{\pm0.35}$ | $8.57_{\pm0.04}$ | $38.17_{\pm2.00}$ |
| | FDR | $30.91_{\pm2.74}$ | $91.01_{\pm0.68}$ | $22.02_{\pm0.08}$ | $61.72_{\pm1.02}$ | $8.70_{\pm0.19}$ | $40.36_{\pm0.68}$ |
| | DER++ | $64.88_{\pm1.17}$ | $91.92_{\pm0.60}$ | $29.60_{\pm1.14}$ | $\underline{62.49}_{\pm1.02}$ | $10.96_{\pm1.17}$ | $40.87_{\pm1.16}$ |
| | Co$^2$L | $\underline{65.57}_{\pm1.37}$ | $93.43_{\pm0.78}$ | $\underline{31.90}_{\pm0.38}$ | $55.02_{\pm0.36}$ | $13.88_{\pm0.40}$ | $42.37_{\pm0.74}$ |
| | TARC | $53.23_{\pm0.10}$ | - | $23.48_{\pm0.10}$ | - | $9.57_{\pm0.12}$ | - |
| | ER-ACE | $62.08_{\pm1.44}$ | $92.20_{\pm0.57}$ | $35.17_{\pm1.17}$ | $63.09_{\pm1.23}$ | $11.25_{\pm0.54}$ | $44.17_{\pm1.02}$ |
| | CLS-ER[1] | $61.88_{\pm2.43}$ | $\underline{93.59}_{\pm0.87}$ | - | - | $\underline{17.68}_{\pm1.65}$ | $\underline{52.60}_{\pm1.56}$ |
| | DRI | $65.16_{\pm1.13}$ | $92.87_{\pm0.71}$ | - | - | $17.58_{\pm1.24}$ | $44.28_{\pm1.37}$ |
| | TAMiL | $\mathbf{68.84}_{\pm1.18}$ | $\mathbf{94.28}_{\pm0.31}$ | $\mathbf{41.43}_{\pm0.75}$ | $\mathbf{71.39}_{\pm0.17}$ | $\mathbf{20.46}_{\pm0.40}$ | $\mathbf{55.44}_{\pm0.52}$ |
| | TAMiL (Oracle) | $\mathbf{91.08}_{\pm0.91}$ | $91.08_{\pm0.91}$ | $71.21_{\pm0.27}$ | $71.68_{\pm0.15}$ | $54.41_{\pm0.49}$ | $55.78_{\pm0.75}$ |
| 500 | ER | $57.74_{\pm0.27}$ | $93.61_{\pm0.27}$ | $28.02_{\pm0.31}$ | $68.23_{\pm0.17}$ | $9.99_{\pm0.29}$ | $48.64_{\pm0.46}$ |
| | FDR | $28.71_{\pm3.23}$ | $93.29_{\pm0.59}$ | $29.19_{\pm0.33}$ | $69.76_{\pm0.51}$ | $10.54_{\pm0.21}$ | $49.88_{\pm0.71}$ |
| | DER++ | $72.70_{\pm1.36}$ | $93.88_{\pm0.50}$ | $\underline{41.40}_{\pm0.96}$ | $\underline{70.61}_{\pm0.08}$ | $19.38_{\pm1.41}$ | $51.91_{\pm0.68}$ |
| | Co$^2$L | $\underline{74.26}_{\pm0.77}$ | $\mathbf{95.90}_{\pm0.26}$ | $39.21_{\pm0.39}$ | $62.98_{\pm0.58}$ | $20.12_{\pm0.42}$ | $53.04_{\pm0.69}$ |
| | TARC | $67.41_{\pm0.41}$ | - | $31.50_{\pm0.40}$ | - | $13.77_{\pm0.17}$ | - |
| | ER-ACE | $68.45_{\pm1.78}$ | $93.47_{\pm1.00}$ | $40.67_{\pm0.06}$ | $66.45_{\pm0.71}$ | $17.73_{\pm0.56}$ | $49.99_{\pm1.51}$ |
| | CLS-ER[1] | $70.40_{\pm1.21}$ | $94.35_{\pm0.38}$ | - | - | $\underline{24.97}_{\pm0.80}$ | $\underline{61.57}_{\pm0.63}$ |
| | DRI | $72.78_{\pm1.44}$ | $93.85_{\pm0.46}$ | - | - | $22.63_{\pm0.81}$ | $52.89_{\pm0.60}$ |
| | TAMiL | $\mathbf{74.45}_{\pm0.27}$ | $\underline{94.61}_{\pm0.19}$ | $\mathbf{50.11}_{\pm0.34}$ | $\mathbf{76.38}_{\pm0.30}$ | $\mathbf{28.48}_{\pm1.50}$ | $\mathbf{64.42}_{\pm0.27}$ |
| | TAMiL (Oracle) | $\mathbf{93.93}_{\pm0.38}$ | $93.93_{\pm0.38}$ | $76.75_{\pm0.12}$ | $76.88_{\pm0.11}$ | $64.06_{\pm2.38}$ | $64.55_{\pm2.14}$ |

[1] Single EMA model.

**Comparison with evolving architectures**: Similar to progressive networks (e.g. PNN, CPG, PAE), dynamic sparse networks (e.g. CLIP, NISPA, PackNet) reduce task interference by learning a non-overlapping task-specific sparse architecture within a fixed capacity model. We consider these two approaches to be two extremes of evolving architectures in CL and present a comparison with TAMiL on Seq-CIFAR100 (20 tasks, buffer size 500) under the Task-IL scenario. Figure 2 presents *final task accuracies* after training on all tasks. Although TAMiL uses a slightly larger model (Appendix D.3), it does not suffer from capacity saturation and retains strong performance compared to fixed capacity models. On the other hand, progressive networks grow in size when encountered with a new task: PNN grows exorbitantly while CPG by 1.5x, PAE by 2x (results taken from Table 1 in Hung et al. (2019a)) and TAMiL by 1.12x (Table 5) for 20 tasks compared to a fixed capacity model. Therefore, TAMiL and CPG grow more slowly than other progressive networks. TAMiL outperforms all progressive networks with an average accuracy of 84% on all 20 tasks. As earlier layers capture task-agnostic information, scalable parameter isolation in the later layers largely benefits TAMiL.

**Effect of task-attention on prior art**: Analogous to our method, we attempt to augment several existing rehearsal-based methods by equipping them with the TAMs. Figure 3(left) provides a comparison of CL methods with and without TAM when trained on Seq-CIFAR100 (5 tasks) with buffer size 500 in the Class-IL scenario. We also provide an ablation of contribution of different components in TAMiL in A.1. Quite evidently, TAMs drastically improve the performance of all CL methods, more so when the true TAM is used for inference (oracle). Independent of the underlying learning mechanism, these dedicated modules admit only the task-relevant information from the common representation space to the global workspace when warranted by a task-specific input, thereby drastically reducing interference. The generalizability of the effectiveness of TAMs across algorithms reinforces our earlier hypothesis that emulating GWT in computational models can greatly benefit CL with systematic generalization across tasks.

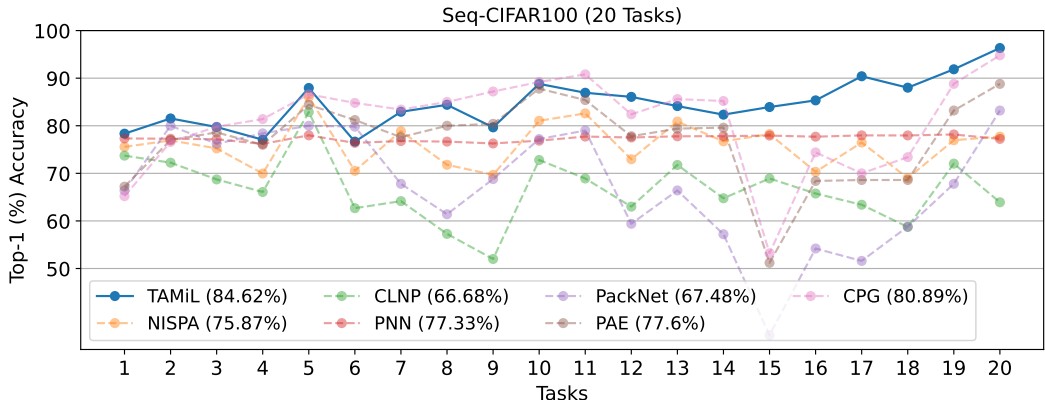

Figure 2: Comparison of final task accuracies of evolving architectures after training on all 20 tasks in Seq-CIFAR100. Mean accuracy on all tasks after training is provided in the legend. TAMiL outperforms all evolving architectures considered in this work.

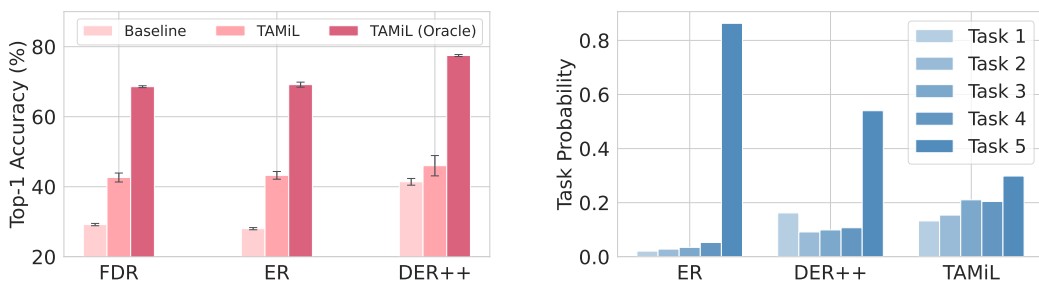

Figure 3: Left: Comparison of top-1 accuracy (%) of CL models with and without TAMs. Right: Average task probabilities of different CL models after Cl training. Both of the above experiments were done on Seq-CIFAR100 with buffer size 500.

**Task-recency bias**: CL models trained in an incremental learning scenario tend to be biased towards the most recent tasks, termed task-recency bias (Hou et al., 2019). Following the analysis of recency bias in Buzzega et al. (2020); Arani et al. (2022), we present the task probabilities in Figure 3 (right). We first compute the prediction probabilities of all samples and average them. For each task, the task probability stands for the sum of average prediction probabilities of the associated classes. The predictions of ER are biased mostly towards recent tasks, with the most recent task being almost 8X as much as the first task. On the contrary, the predictions in TAMiL are more evenly distributed than the baselines, greatly mitigating the task recency bias.

**Performance under longer-task sequences**: Computational systems deployed in the real world are often exposed to a large number of sequential tasks. For rehearsal-based methods with a fixed memory budget, the number of samples in the buffer representing each previous task is drastically reduced in longer sequences of tasks, resulting in poor performance, called long-term catastrophic forgetting (Peng et al., 2021). Therefore, it is quintessential for the CL model to perform well under low buffer regimes and longer task sequences. Figure 4 provides an overview of the performance of CL models with 5, 10, and 20 task sequences on Seq-CIFAR100 with a fixed buffer size of 500. As the number of tasks increases, the number of samples per class decreases, resulting in increased forgetting. Our method equipped with TAMs preserves the previous task information better and exhibits superior performance over baselines even under extreme low-buffer regimes.

**Model calibration**: A well-calibrated model improves reliability by reducing the expectation difference between confidence and accuracy (Guo et al., 2017). Figure 5 shows the Expected Calibration Error (ECE) along with a reliability diagram on Seq-CIFAR100 using a calibration framework (Kuppers et al., 2020). As can be seen, ER is highly miscalibrated and more overconfident than other CL

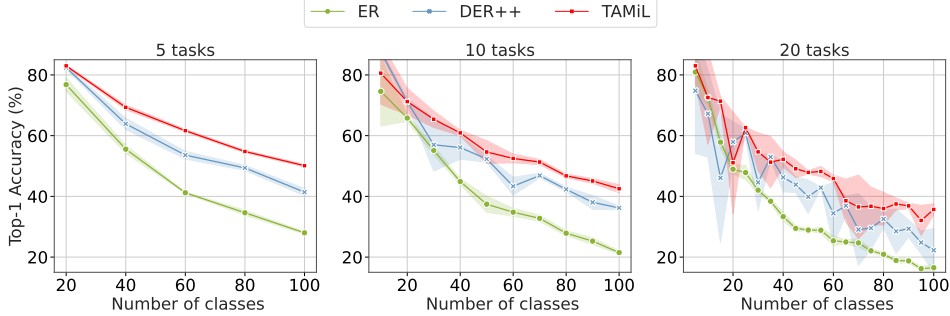

Figure 4: Comparison of Top-1 accuracy (%) of CL models in Seq-CIFAR100 with different number of tasks. TAMiL consistently outperforms the baselines under longer task sequences.

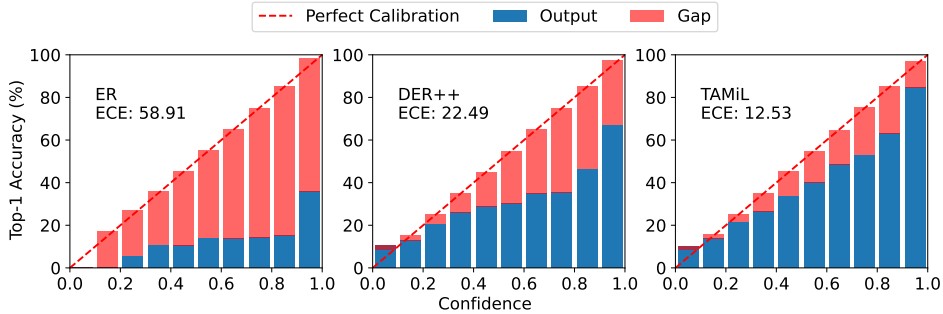

Figure 5: Reliability diagram along with ECE for different CL models trained on Seq-CIFAR100 (buffer size 500) with 5 tasks. TAMiL is well-calibrated when compared to the baselines.

models. On the other hand, TAMiL has the lowest ECE by ensuring that the predicted softmax scores are better indicators of the actual probability of a correct prediction. In addition to drastically reducing catastrophic forgetting in CL, TAMs in our approach help mitigate miscalibration.

## 6 CONCLUSION

We proposed TAMiL, a novel CL approach that encompasses both experience rehearsal and self-regulated scalable neurogenesis to further mitigate catastrophic forgetting in CL. Inspired by the Global Workspace theory (GWT) of conscious information access in the brain, Task-specific Attention Modules (TAMs) in our approach capture task-specific information from the common representation space, thus greatly reducing task interference. The generalizability of the effectiveness of TAMs across CL algorithms reinforces the applicability of GWT in computational models in CL. Given the bottleneck nature of TAMs, the additional parameters introduced in each task are negligible in size compared to parameter growth in PNNs. TAMiL neither suffers from capacity saturation nor scalability issues and retains strong performance even when exposed to a large number of tasks. Although TAMiL performs extremely well, more sophisticated matching criteria can be developed to shore up performance close to the oracle version in the future.

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

# A  ANALYSIS OF TASK-SPECIFIC ATTENTION MODULES

## A.1  ABLATION STUDY

We attempt to disentangle the contribution of key components in our approach. Table 2 provides an ablation study of our method trained on Seq-CIFAR100 with buffer size 500 for 5 tasks (More ablation on TAMs can be found in Appendix A.3). When the EMA model is absent, we store past predictions in the buffer for consistency regularization. As can be seen, each component contributes significantly to the overall performance of TAMiL

Table 2: Ablations of the different key components of our proposed method. The Top-1 accuracy (%) is reported on Seq-CIFAR100 for the 500 buffer size learned with 5 tasks.

| EMA Model | Pairwise loss | TAMs | Class-IL | Task-IL |
|:---:|:---:|:---:|:---:|:---:|
| ✓ | ✓ | ✓ | $\mathbf{50.11}_{\pm0.34}$ | $\mathbf{76.47}_{\pm0.51}$ |
| ✗ | ✓ | ✓ | $47.51_{\pm0.96}$ | $73.79_{\pm0.51}$ |
| ✗ | ✗ | ✓ | $45.10_{\pm3.46}$ | $73.34_{\pm0.67}$ |
| ✗ | ✗ | ✗ | $41.40_{\pm0.96}$ | $70.61_{\pm0.08}$ |

## A.2  SELECTION OF APPROPRIATE TAM DURING INFERENCE

Continual learning in the brain is mediated by a rich set of neurophysiological processes that harbor different types of knowledge, and conscious processing integrates them coherently. The Global Workspace Theory (GWT) (Baars, 1994) of the conscious information access in the brain states that only the behaviorally relevant information from the perceptual contents in the common representation space are admitted to the global workspace when warranted by a task. However, unless there is sufficient activation in the prefrontal region, information is not always consciously processed in the brain (Juliani et al., 2022). To this end, Global Neuronal Workspace (GNW) hypothesis (Dehaene et al., 1998) posits that brain entails a second computational space composed of widely distributed excitatory neurons that selectively mobilize or suppress, through descending connections, the contribution of specific processor neuron. GNW acts as a router associated with the different brain regions through which the information is selected and made available when triggered by an external stimulus Mashour et al. (2020).

GNW is associated with an *ignition* event (Dehaene et al., 2003) characterized by the activation of subset of workspace neurons and inhibition of the rest of the neurons. Analogously, the TAMs in our network $\tau_\theta = \{\tau_\theta^k \mid k \leq t\}$ act as a communication bottleneck and are associated with an ignition event defined in Equation 5. Although quite simple in its formulation, Equation 5 activates a subset of neurons (an appropriate TAM) and inhibits the rest of neurons (rest of the TAMs) from processing the incoming information. When warranted by a task-specific input, the gating mechanism in Equation 5 allows only relevant information to pass through the global workspace. The appropriate activation and inhibition of TAMs is quintessential for reducing interference between tasks. As is clear from the experimental evaluation in Table 1, any deviation from *Oracle* results in higher forgetting. More complex alternatives such as learning a policy using reinforcement learning, a gating mechanism using Gumbel-softmax, and prototype matching can also be explored in place of the proposed ignition event to further improve the selection accuracy.

## A.3  CHOICE OF TAMs

The prefrontal cortex of the primate brain is presumed to have task-dependent neural representations that act as a gating in different brain functions (Mante et al., 2013). When warranted by a task-specific input, the gating mechanism allows only the relevant information to pass through the global workspace. As noted in Section 3.2, emulating such task-specific attention modules in computational systems comes with several design constraints, including scalability, effectiveness, etc. Table 3 shows some of the TAMs considered in this work. The undercomplete autoencoder (the encoder learns a lower-dimensional embedding than input layer) with ReLu non-linearity as opposed to multi-layer perceptron (MLP) or linear layer achieves the best performance. A linear autoencoder with a Euclidean loss function learns the same subspace as PCA. However, AE with nonlinear

Table 3: Ablation of the different types of task-attention in place of TAMs in our proposed method. The accuracy is reported on Seq-CIFAR100 for the 500 buffer size learned with 5 tasks.

| TAMs | Output non-linearity | Top-1 (%) Seq-CIFAR100 Class-IL |
|---|---|---|
| Linear layer | - | 41.96 ±2.32 |
| No learnable layer | Sigmoid | 42.04 ±0.49 |
| Multi-layer Perceptron | Sigmoid | 46.08 ±4.99 |
| Autoencoder | ReLu | 44.31 ±0.18 |
| | Tanh | 40.78 ±1.56 |
| | Sigmoid | **49.01** ±1.11 |

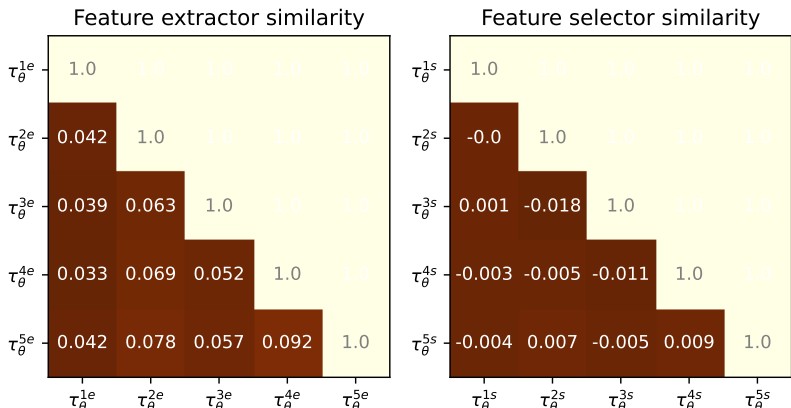

Figure 6: Cosine similarity between different weight feature extractors and selectors of different TAMs under Seq-CIFAR100 (buffer size 500). Although each TAM receives the same input from the common representation space, each TAM learns a different embedding resulting in different attention for each task. Therefore, the cosine similarity between any two TAM is negligibly small.

functions yields better dimensionality reduction compared to PCA (Hinton & Salakhutdinov, 2006). Therefore, in our proposed approach, we chose autoencoder with ReLu non-linearity in the latent stage and sigmoid activation in the output stage as TAM.

## A.4   TAMs SIMILARITY

We attempt to improve the understanding of TAMs in our proposed method. Each of these TAMs consists of two parts $\tau_\theta^i = \{\tau_\theta^{ie}, \tau_\theta^{is}\}$, where $\tau_\theta^{ie}$ acts as a feature extractor and $\tau_\theta^{is}$ as a feature selector. The feature extractor learns a low-dimensional subspace using a linear layer followed by ReLU activation. On the other hand, the feature selector learns task-specific attention using another linear layer followed by sigmoid activation. When using task-specific attention in Class-IL / Task-IL, one would envisage TAMs to capture drastically different information for each task, as each task in Class-IL / Task-IL is vastly different. As the knowledge of the learned tasks is encoded in the weights (Krishnan et al., 2019), we envisage to compute the similarity between weight matrices to gauge whether TAMs are indeed capturing different information. As cosine similarity is widely used in high-dimensional spaces (Luo et al., 2018), we plot the cosine similarity between respective

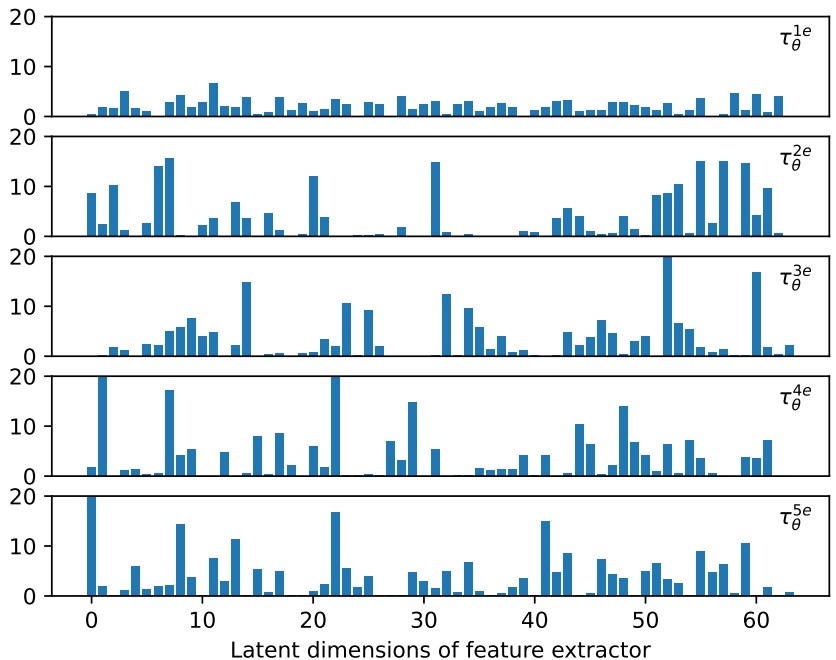

Figure 7: Average activation of feature extractors within each TAM on Seq-CIFAR100. As can be seen, each TAM maps the common representation space to a different latent space thereby reducing the interference. We attribute this behaviour to pairwise discrepancy enforced through Equation 4.

feature extractor and selector weight matrices of each TAMs in Figure 6. As can be seen, all TAMs are vastly different from each other inline with the tasks they were exposed to earlier. We attribute this functional diversity to the pairwise discrepancy loss described in Section 3.2 and Equation 4). As evident in Figure 7, the average activation of the feature extractors begins to diverge from Task-2 as pairwise discrepancy loss kicks in. From Task-2, the average activations are coherent, distributed, and diverse from each other. Due to the limited size of the embedding dimension, there is sparsity in activations in a desirable byproduct.

### A.5 Parameter growth comparison

We compare the parameter growth in TAMiL with respect to fixed-capacity models and PNNs. Table 4 presents a comparison of parameter growth in 5, 10, and 20 tasks. As the EMA model is not central to the working of our method, we present two versions of TAMiL with and without the EMA model. Compared to a fixed capacity model, TAMiL (without EMA) grows only marginally at 11% even for 20 tasks. Having an EMA model doubles the parameter growth, as both EMA and working model will have same number of TAMs. On the other hand, the number of parameters in PNNs grows exponentially with the number of tasks, thus rendering them inapplicable in real-world scenarios. As shown earlier in Section 5, TAMiL neither suffers from capacity saturation nor from scalability issues, thus producing strong performance even in longer task sequences.

## B Limitations

Inspired by the GWT, we propose TAMiL, a continual learning method that entails task attention modules to capture task-specific information from the common representation space. Although TAMiL performs extremely well on different CL scenarios, it is not without limitations: TAMiL assumes that the common representation space captures the information generalizable across tasks.

Table 4: Growth in number of parameters for different number of task sequences.

| Methods | Number of parameters (Millions) | | |
|---|---|---|---|
| | 5 tasks | 10 tasks | 20 tasks |
| Fixed capacity model | 11.23 | 11.23 | 11.23 |
| TAMiL (without EMA) | 11.55 | 11.88 | 12.54 |
| TAMiL (with EMA) | 23.10 | 23.76 | 25.08 |
| PNNs | 297.21 | 874.01 | 2645.05 |

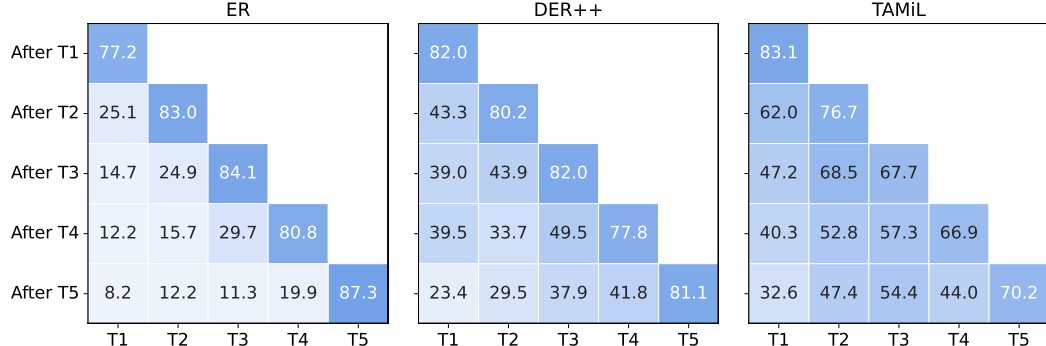

Figure 8: Task-wise performance of CL models trained on Seq-CIFAR100 with buffer size 500 on 5 tasks. The performance of the baseline models is mostly emanating from the performance on the last task while TAMiL achieves considerably more distributed performance on all tasks.

Violation of this assumption limits the ability of TAMs to capture task-specific information. Second, TAMiL requires task boundary information to switch to a new TAM to avoid interference between tasks. We plan to leverage task similarity to merge multiple TAMs into one to avoid this constraint in the future. Finally, as is clear from the experimental evaluation in Table 1, any deviation from *Oracle* results in higher forgetting. TAMiL can benefit from a more accurate matching criterion to match the performance of Oracle. More complex alternatives such as learning a policy using reinforcement learning, a gating mechanism using Gumbel-softmax, and prototype matching can also be explored in place of the proposed matching criterion to further improve the selection accuracy.

## C  TASK PERFORMANCE

### C.1  TASK-WISE PERFORMANCE

In Table 1, we report the final accuracy after learning all tasks in the Class-IL and Task-IL scenarios. In Figure 8, we disentangle the task-wise performance of different CL models trained on Seq-CIFAR100 with buffer size 500 on 5 tasks. Our proposed method TAMiL retains the performance on previous tasks while the baseline models adapt mostly towards the recent tasks. Therefore, the final average accuracy alone can sometimes be quite misleading.

### C.2  FORGETTING

While one can argue that learning to classify unseen classes is desirable, Class-IL and Task-IL show different classes in different tasks, making transfer impossible (Buzzega et al., 2020). On the contrary, forgetting is an important measure to gauge the overall performance of the CL model. We compute forgetting as the difference between the current accuracy and its best value for each task. Table 5 presents the forgetting results complementary to the results reported in Table 1. As noted earlier, TAMiL drastically reduces forgetting, thereby enabling efficient CL with systematic generalization.

Table 5: Forgetting in CL models across various CL scenarios. The results are the average of 3 runs.

| Buffer size | Methods | Seq-CIFAR10 | | Seq-TinyImageNet | |
|---|---|---|---|---|---|
| | | Class-IL | Task-IL | Class-IL | Task-IL |
| 200 | ER | $61.24_{\pm2.62}$ | $7.08_{\pm0.64}$ | $76.37_{\pm0.53}$ | $43.14_{\pm0.97}$ |
| | DER++ | $32.59_{\pm2.32}$ | $5.16_{\pm0.21}$ | $72.74_{\pm0.56}$ | $41.63_{\pm1.13}$ |
| | TAMiL | $\mathbf{22.03}_{\pm1.89}$ | $\mathbf{3.45}_{\pm0.39}$ | $\mathbf{55.69}_{\pm1.45}$ | $\mathbf{24.54}_{\pm0.94}$ |
| 500 | ER | $45.35_{\pm0.07}$ | $3.54_{\pm0.35}$ | $75.27_{\pm0.17}$ | $31.36_{\pm0.27}$ |
| | DER++ | $22.38_{\pm4.41}$ | $4.66_{\pm1.15}$ | $64.58_{\pm2.01}$ | $27.85_{\pm0.51}$ |
| | TAMiL | $\mathbf{15.95}_{\pm0.86}$ | $\mathbf{3.14}_{\pm0.42}$ | $\mathbf{43.43}_{\pm2.24}$ | $\mathbf{15.68}_{\pm0.18}$ |

Table 6: Comparison of CL models on Seq-CIFAR-100 (5 tasks, with 500 buffer size) in three different settings: Single-head, multi-head, and single-head with TAMs. TAMs and multi-head perform comparably with TAMs producing slightly higher Task-IL performance.

| Methods | Single head | Multi-head (V1) | Multi-head (V2) | With TAMs (Single head) |
|---|---|---|---|---|
| ER | $68.23_{\pm0.17}$ | $68.15_{\pm0.31}$ | $68.60_{\pm0.82}$ | $\mathbf{69.15}_{\pm0.72}$ |
| DER++ | $70.61_{\pm0.08}$ | $75.58_{\pm0.30}$ | $75.77_{\pm0.48}$ | $\mathbf{77.47}_{\pm0.28}$ |
| CLS-ER | $76.00_{\pm0.96}$ | $79.58_{\pm0.38}$ | $78.59_{\pm0.48}$ | $\mathbf{79.62}_{\pm0.11}$ |

## C.3 PERFORMANCE ON SEQ-CORE50

Table 7 provides a comparison of different Cl models on Seq-core50. Following Maltoni & Lomonaco (2019), Seq-Core50 is organized into nine tasks, the first of which includes ten classes, while the rest have five classes each. As can be seen, TAMiL improves performance in most settings. In the oracle version, using a task-specific TAM greatly improves performance, up to 30% in the Class-IL scenario.

## C.4 COMPARISON WITH MULTI-HEAD SETUP

We seek to provide an understanding of how task-specific parameters improve learning in sequential tasks. Table 6 describes an ablation of three baseline methods, namely ER, DER++, and CLS-ER (single EMA model version) in the presence of multiple heads and TAMs in the Task-IL setting. We report the results on Seq-CIFAR100 with 5 tasks and a buffer size of 500. Each method was evaluated under single-head setting, multi-head setting, and single-head with TAMs setting. We tried two variants within multihead setting: multihead (v1) has a linear layer for each task representing classes within respective tasks, while multihead (v2) has a two-layer MLP for each task with number of parameters comparable with TAMs. As per the original formulation of CLS-ER, we evaluated the EMA model instead of the working model.

s can be seen, TAMs and multi-head (both versions) perform comparably with TAMs producing slightly higher Task-IL performance. As is clear from the multi-head versions, having more task-specific parameters does not necessarily result in significant improvement. On the other hand, TAMs offer an alternative way of augmenting CL models with task-specific parameters. Besides, TAMs are much more than plain task-specific parameters: with an appropriate ignition event, TAMs can be easily adapted to the Class-IL setting without requiring task identity at inference. In the future, a comprehensive method that includes both multiple-heads and TAMs can be developed to further improve the performance of CL models.

It is important to note that multiple heads in Task-IL bring limited performance improvement compared to single head. However, multiple heads require task identity at inference time and do not work in Class-IL scenario. This is also true for several progressive networks (PNN, CPG, and PAE) considered in this work. This limits their applicability to real-life scenarios. On the other hand, TAMiL performs extremely well in both Class-IL and Task-IL without having to change anything in the proposed method. Therefore, we argue that TAMs bring more sophistication and design flexibility than their task-specific counterparts in other approaches.

Table 7: Comparison of CL models on Seq-core50. We provide the average Top-1 (%) accuracy of all tasks after CL training.

| Buffer size | Methods | Seq-Core50 | |
| --- | --- | --- | --- |
| | | Class-IL | Task-IL |
| 200 | ER | $21.49_{\pm 0.56}$ | $65.63_{\pm 0.92}$ |
| | DER++ | $28.47_{\pm 0.61}$ | $68.50_{\pm 1.03}$ |
| | TAMiL | $\mathbf{32.67}_{\pm 0.36}$ | $\mathbf{70.76}_{\pm 1.05}$ |
| | TAMiL (Oracle) | $\mathbf{64.04}_{\pm 1.34}$ | $70.53_{\pm 0.47}$ |
| 500 | ER | $29.39_{\pm 0.77}$ | $69.90_{\pm 0.95}$ |
| | DER++ | $\mathbf{40.31}_{\pm 1.49}$ | $75.94_{\pm 0.24}$ |
| | TAMiL | $39.15_{\pm 1.48}$ | $\mathbf{77.36}_{\pm 0.60}$ |
| | TAMiL (Oracle) | $\mathbf{70.90}_{\pm 0.43}$ | $76.53_{\pm 0.20}$ |

---

**Algorithm 2** Reservoir sampling (Vitter, 1985)

---

    **input:** Data streams $\mathcal{D}_t, \mathcal{D}_m, \{x, y\} \in \mathcal{D}_t$
    Maximum buffer size $\mathcal{M}$, current buffer size $\mathcal{N}$
1: **if** $\mathcal{M} > \mathcal{N}$ **then**
2:     $\mathcal{D}_m[\mathcal{N}] \leftarrow \{x, y\}$
3: **else**
4:     $v = randomInteger(min = 0, max = \mathcal{N})$
5:     **if** $v < \mathcal{N}$ **then**
6:         $\mathcal{D}_m[v] \leftarrow \{x, y\}$
7: **return** $\mathcal{D}_m$

---

# D   IMPLEMENTATION DETAILS

## D.1   RESERVOIR SAMPLING

Algorithm 2 describes the steps for building a memory buffer using the reservoir sampling strategy (Vitter, 1985). Reservoir sampling assigns equal probability to each sample of a data stream of unknown length to be represented in the memory buffer. When the buffer is full, the replacements are made randomly.

## D.2   DATASETS AND MODEL

We obtain Seq-CIFAR10, Seq-CIFAR100 and Seq-TinyImageNet by splitting CIFAR10 (Krizhevsky et al., 2009), CIFAR100 (Krizhevsky et al., 2009) and TinyImageNet (Le & Yang, 2015) into 5, 5 and 10 partitions of 2, 20 and 20 classes per task, respectively. We also experiment with longer task sequences in Seq-CIFAR100 by increasing number of tasks to 5, 10 and 20 while correspondingly decreasing number classes to 20, 10, and 5, respectively. Following Arani et al. (2022); Buzzega et al. (2020); Cha et al. (2021); Caccia et al. (2021a), we employ ResNet-18 (He et al., 2016) as the backbone to learn a common representation space for all our experiments. We use a single, expanding linear classifier representing all classes belonging to all tasks. The training regime for both Class-IL and Task-IL are as follows: The CL model is trained on all tasks sequentially with/without experience-rehearsal with reservoir sampling depending on its formulation. During training, entire network is updated including the linear classifier. The training scheme is same for both Class-IL and Task-IL. For comparison between different state-of-the-art methods, we report the average of accuracies on all tasks seen so far in Class-IL. As is the standard practice in Task-IL, we leverage the task identity and mask the neurons that do not belong to the prompted task in the linear classifier.

Our CL model consists of as many TAMs as the number of tasks. Each TAM is an autoencoder with a linear layer with ReLu activation as an encoder and a linear layer with sigmoid activation as a decoder. Both input and output are 512 dimensional while the latent space is of 64 dimensions. As TAMiL involves an EMA model for consistency regularization, we use CLS-ER's two-model version with a single EMA copy for a fair comparison. As TAMs can be plugged into any rehearsal-based approach, we plan to improve multiple-EMA CLS-ER with TAMs in the future.

### D.3 BACKBONES USED FOR COMPARISON WITH DYNAMIC SPARSE NETWORKS

Diverging from the mainstream practice of utilizing a dense CL model, dynamic sparse approaches start with a sparse network and maintain the same connection density throughout the learning trajectory. As sparsifying a CL model involves disentangling interfering units to avoid forgetting and creating novel pathways to encode new knowledge, implementing batch normalization and residual connections is not trivial for both NISPA and CLNP. Therefore, these methods do not use the ResNet-18 architecture. Instead, they opt for a simple CNN architecture without skip connections and batch normalization. On the other hand, TAMiL is not prone to complexities in the underlying model and is therefore simple to plug-and-play for any approach with any kind of backbone.

### D.4 MAINTAINING AN EMA MODEL FOR CONSISTENCY REGULARIZATION

Knowledge of previous tasks can be better preserved using consistency regularization in CL (Bhat et al., 2022a). To enforce consistency, the previous predictions can be stored along with the image in the buffer or an EMA teacher model can be employed to distill the knowledge of the previous tasks. In DER++, previous predictions are stored in the buffer. In Figure 3(left) we plug-and-play TAMs on top of DER++ and show discernible improvement, indicating that the effectiveness of TAMs is independent of the use of EMA model.

The EMA of a model can be considered as forming a self-ensemble of the intermediate model states that leads to better internal representations (Arani et al., 2022). Therefore, using an EMA model instead of storing the logits yields better results in CL. Therefore, we use an EMA model in all our experiments in Table 1. When training a CL model in TAMiL, we stochastically update the EMA model as follows:

$$\theta_{EMA} = \begin{cases} \theta_{EMA}, & \text{if } \gamma \leq \mathcal{U}(0,1) \\ \eta\,\theta_{EMA} + (1-\eta)\,\theta, & \text{otherwise} \end{cases} \tag{7}$$

where $\eta$ is a decay parameter, $\gamma$ is a update rate and, $\theta$ and $\theta_{EMA}$ represent weights of CL model and EMA model respectively. During each iteration, buffered input is passed through each of these models and CL model's predictions are enforced to be consistent with the EMA model's predictions.

### D.5 INTUITION BEHIND WORKING OF IGNITION EVENT

The TAMs in our framework act as a communication bottleneck and select features relevant for the corresponding task. However, association between an input sample and its corresponding TAM is not given as task identity is not available during inference in Class-IL. Motivated by ignition event in the brain (Appendix A.2), we develop a simple ignition event to select appropriate TAM both during training and inference. To this end, during training, each TAM first learns task-specific attention using task-specific data $\mathcal{D}_t$. As our method employs experience-rehearsal, we use $\mathcal{D}_m$ to automatically select the appropriate TAM. Since each TAM is associated with a specific task, inferring a wrong TAM for the buffered samples can result in sub-par performance and higher penalty in terms of cross-entropy loss and consistency regularization. This way, the CL model is trained to first capture task-specific information and also learn the routing through buffered samples using an ignition event.

CL models without TAMs (DER++, CLS-ER), already accumulate information in their common representation space that is sufficient for decent classification performance. TAMs, on the other hand, denoisify these features resulting in higher performance due to lessened interference. We empirically found that deviating too much from common representation space features incurred higher interference and consequent forgetting in presence of TAMs. Therefore, the task-specific attention should be such that it promotes denoising, but not at the expense of features important for the current task. To this end, we proposed a simple matching criterion that dynamically selects a TAM

that is most similar to common representation space features. For buffered samples, appropriate TAM is dynamically selected using Equation 5. Only the output of selected TAM is forward propagated to global workspace. We then compute cross entropy loss and consistency regularization, and backpropagate the errors.

The obvious downside of such an approximation is a drop in performance. Compared to Oracle version, TAMiL with ignition event described in Equation 5 produces a subpar performance. We note this obvious limitation in Appendix B. TAMiL can benefit from more accurate matching criterion to match the performance of Oracle. More complex alternatives such as learning a policy using reinforcement learning, a gating mechanism using Gumbel-softmax, and prototype matching can also be explored in place of proposed matching criterion to further improve the selection accuracy.

