# OpenReview forum: "Task-Aware Information Routing from Common Representation Space in Lifelong Learning"
_ICLR.cc/2023/Conference — ICLR 2023 poster_

### Official Review · Reviewer_HdW5 · 2022-10-24

**Confidence:** 5
**Correctness:** 3
**Technical Novelty And Significance:** 2
**Empirical Novelty And Significance:** 3
**Recommendation:** 6

**Clarity, Quality, Novelty And Reproducibility:**

The main strength of this submission is, to me, the thoroughness of the empirical evaluation in terms of the number of analyses and insights from the data. I wish more papers took their experimental design as seriously as this one does. Kudos to the authors for that.

My biggest concern with this submission is broadly speaking its placement with respect to the existing literature. In particular:
1. There seem to be a number of missing citations that are highly relevant to the submission. Here are some existing threads and accompanying citations that I believe should be discussed in a revised manuscript
	- Multiple works have proposed approaches to add task-specific parameters in ways that are far more space efficient than PNNs. The common technique is to use task-specific output classification heads, but other approaches have been proposed (e.g., [1,2])
	- Other works have proposed methods to automatically expand the capacity of a NN in a way that grows drastically slower than PNNs, but does not suffer from capacity saturation (nor does it require explicitly adding new parameters for every task; e.g., [3,4])
	- A particularly relevant line of work, especially given the GWT motivation, seems to be modular continual learning, most of which also allows the agent to automatically grow its capacity automatically (e.g., [5, 6, 7])
	- There are also a number of techniques for automatically determining the task at inference time. [7] contains one example approach, but there are countless others.
I would encourage the authors to discuss their work in light of these multiple threads of related work. In particular, looking at the related literature, my take is that the main three contributions of this submission are: a) the motivation in terms of the GWT, b) the new method for adding task-specific parameters via undercomplete autoencoders, and c) the method for inferring the correct task at test time. Would the authors agree with that statement?
2. The choice of baselines seems somewhat odd, given that the main technical contributions, as I understand, are (b) and (c) in my previous comment. The evaluation should center on the comparison to the various mechanisms for expanding the network capacity (b) and to the techniques for inferring the correct task ID (c). Given the details in the paper, I am unsure whether the competing baselines (especially, the replay-based mechanisms) are allowed to use task-specific output heads. This would be especially fair in the task-incremental setting, where the agent knows which task to assign to each data point. Moreover, I would be interested in seeing some ablation that looks at the difference in performance in TAMiL if instead of using the proposed autoencoder, it used something simpler like task-specific classification heads at the output. I _think_ a version of this is shown in the Appendix in Table 3, but since it was never mentioned in the main paper and I couldn't find enough details in the Appendix describing the ablation, I couldn't be sure.

One concern in terms of the experiments is whether the replay-based baselines were allowed any number of task-specific parameters, even if just task-specific output heads, as mentioned above. If that is not the case, it is possible that the lack of task-specific parameters would account for most of the difference in performance between TAMiL and the baselines. I believe that the baselines should be allowed task-specific parameters, at the very least in the task-incremental setting.

I was also left somewhat confused about exactly how task inference is being learned during the training process. Why is reconstruction error used to choose the appropriate TAM? My understanding was that TAMs were explicitly trained to only pass relevant information, so why would reconstruction error work here? Could the authors clarify the process for training the task inference, right after Eq. 5? How does this process encourage the model to choose the correct TAM? What parameters are being backpropped in this stage? The representation, the chosen TAM, the prediction portion of the net? And why?


############## Additional feedback ##############

The following points are provided as feedback to hopefully help better shape the submitted manuscript, but did not impact my recommendation in a major way.

Would it perhaps be better to not use the term attention to refer to the task-specific modules? Recently, attention has been more and more associated to the specific form of key-query-value attention from transformers, and this is a different type of attention. Or, maybe the term can be kept, but the manuscript might clarify earlier on what attention means in this particular context and how it's unrelated to attention in transformer models.

Sec 3
- The loss in Eq 4 induces another linear dependency on the number of previous tasks. Is this done over each task's data, the new data, or the union of replay and new data? Based on the description around Eq 5, I'd imagine it's only new task data
- "In addition to Lpd (Eq. 4), we do not use any other objective on the TAMs to constrain their learning" -- so there is no loss term for the autoencoder to reconstruct its input? Then are these really autoencoders? It's just semantics, but shouldn't autoencoders be trained to reconstruct their inputs? Perhaps just stating that they have a similar structure to autoencoders but are not trained for reconstruction might be a good idea.

Sec 5
- I liked the study in Fig. 3 left. Did the authors look at how this compares to other forms of adding task-specific parameters, such as calibration parameters or task-specific classification heads, in terms of number of additional parameters and performance? It seems that this might be included in Table 3 in the appendices, but it was never mentioned in the paper and I couldn't find enough details in the appendix. Could the authors please expand on this?
- Could the authors describe in a bit more detail what the exact experiment in Fig. 3 right is? What exactly is the task probability?
- What's the intent of Fig. 4? While clearly TAMiL outperforms the baselines (which was already established), the gap between them is actually fairly constant through the three task-sequence lengths. This suggests that all three methods under comparison are affected roughly equivalently by the length of the task sequence. What insight should we gain from this result?
- Table 2 ablates the use of the exponential moving average, but what is the reasoning behind using that instead of storing predictions, and why should we expect it to perform better (as it clearly does)?

Supplement
- The submission mentions that code will be released, but it is not included for review.

Typos/style/grammar/layout
- "one such knowledge bases is"
- It might be best to place all tables/figures at the top or bottom of the page, though I don't believe the formatting instructions actually request this.

[1] Singh et al. "Calibrating CNNs for Lifelong Learning". NeurIPS 2020
[2] Ke et al. "Achieving forgetting prevention and knowledge transfer in continual learning". NeurIPS 2021
[3] Yoon et al. "Lifelong Learning with Dynamically Expandable Networks". ICLR 2018
[4] Hung et al. "Compacting, Picking and Growing for Unforgetting Continual Learning". NeurIPS 2019
[5] Mendez et al. "Lifelong Learning of Compositional Structures". ICLR 2021
[6] Veniat et al. "Efficient Continual Learning with Modular Networks and Task-Driven Priors". ICLR 2021
[7] Ostapenko et al. "Continual Learning via Local Module Composition". NeurIPS 2021


**Strength And Weaknesses:**

########### Strengths ###########
- The experimental evaluation is notably thorough, including a variety of analyses and ablations that seek to understand various aspects of the improvement of TAMiL with respect to the chosen baselines
- The idea to use feature selectors as an alternative to classification heads as an alternative form of task-specific parameters is interesting and warrants further investigation

########### Weaknesses ###########
- There are a number of references that should likely be included and compared against qualitatively (at least) or quantitatively (ideally)
- Some details about the experimental evaluation are missing, which makes me wonder about the fairness of the comparisons
- I wasn't able to follow exactly how the approach automatically infers the task ID during evaluation, and I believe that additional explanation would be needed


**Summary Of The Paper:**

The submission proposes TAMiL, a continual learning method inspired by aspects of the global workspace theory that trains task-specific attention modules to select which features from a shared representation are relevant to the current task. The approach uses replay in combination with functional regularization to avoid forgetting, and trains the system in such a way that it can automatically determine the correct task at inference time in class-incremental settings. The results demonstrate that TAMiL outperforms the chosen baselines in three standard CL benchmarks.


**Summary Of The Review:**

This submission introduces a new approach for continual learning inspired by the GWT which uses task-specific attention modules to specialize a shared representation to each individual task, and automatically detects tasks at inference time using these modules. The experimental evaluation is quite thorough in terms of analyzing the results with various perspectives and running a few ablative tests. My biggest concerns are the placement with respect to the literature, both in presenting the novelty of the approach and in comparing against existing methods empirically, and potential lack of fairness in the evaluations. Due to these concerns, I am leaning towards recommending the rejection of this work, but encourage the authors to engage in the discussion.

############# Update after rebuttal #############

I am increasing my score from 5 (marginally below threshold) to 6 (marginally above threshold) per the discussion with the authors below.

---

> ### Author Response · Authors · 2022-11-11
> **Reply to the reviewer  HdW5**
>
> >The choice of baselines seems somewhat odd,
>
> We compare with the state-of-the-art rehearsal-based methods and parameter-isolation (evolving architectures) methods. The relevance of each of these related works is as follows: TAMiL employs rehearsal with reservoir sampling with a single expanding head representing all classes in both Class-IL and Task-IL settings. Therefore, we compare TAMiL with several state-of-the-art rehearsal-based methods. We keep the training regime (e.g., training schedule, reservoir sampling, ResNet-18 as backbone, single head etc.) same to make comparison fair between these approaches.  In addition, as TAMiL involves task-specific attention modules, we consider parameter isolation methods for comparison. Among parameter isolation, dynamic sparse networks work within a fixed model capacity while progressive networks expand model capacity either naively or intelligently using some criterion. We compare TAMiL  with both of these approaches to provide a bigger picture.
> We respectfully disagree with the reviewer that the performance improvement is solely arising from additional number of parameters. Table 3 in Appendix A.2 describes an ablation of different kinds of attention in place of proposed TAMs. This describes the prominence of proposed TAM and reasons behind its choice. We also updated the Ablation study in Section 5 to refer to this section in the Appendix for more information on TAMs.
>
> >One concern in terms of the experiments is whether the replay-based baselines were allowed any number of task-specific parameters,
>
> In Table 1 all rehearsal-based approaches have been compared in their original formulation (i.e. single head representing all classes). Only PNN was allowed to have task specific parameters. We respectfully disagree with the reviewer that the performance improvement is solely arising from an additional number of parameters. Table 3 already showcases that other alternatives for TAMs with same number of parameters do not work as well as the proposed method. As per the reviewer’s suggestion, we conduct a  study with three baseline CL approaches with multiple heads in Appendix D.4 and show that TAMs indeed bring discernible benefits in CL.
>
> >Would it perhaps be better to not use the term attention to refer to the task-specific modules
>
> We agree that it can be confusing. We have added a clarification with respect to attention in vision transformers in the Introduction.
>
> >The loss in Eq 4 induces another linear dependency on the number of previous tasks
>
> It is applied only on the current task data.
>
> >"In addition to Lpd (Eq. 4), we do not use any other objective on the TAMs to constrain their learning"
>
> We have now clarified this  with a footnote in Section 3.3.
>
> >I liked the study in Fig. 3 left. Did the authors look at how this compares to other forms of adding task-specific parameters...
>
> We thank the reviewer for appreciation and insightful suggestion. We provide an ablation of different forms of task-attention modules in place of the proposed TAM. We added a reference to Table 3 in the main ablation study in Section 5. We agree that numerous choices for TAMs are possible given the flexibility of TAMiL. We will investigate other possibilities such as calibration modules in the future.
>
> >Could the authors describe in a bit more detail what the exact experiment in Fig. 3 right is? ..
>
> We have updated the Task-recency bias subsection to provide more understanding. Specifically, we first compute the prediction probabilities (softmax of the CL model’s output) of all samples and average them. For each task, task probability stands for sum of average prediction probabilities of associated classes.
>
> >What's the intent of Fig. 4? ...
>
> As described in the paper, rehearsal-based approaches rely heavily on the stored samples to approximate past behavior. However, buffer size is constrained under longer-task sequences as number of samples per class/task reduces overtime. Compared to baselines, TAMiL suffers less from forgetting even under such extreme low-buffer regimes. This shows the practical applicability of TAMiL in scenarios where having a large buffer size is a luxury.
>
> >Table 2 ablates the use of the exponential moving average....
>
> Appendix E.4 in the updated manuscript provides reasoning behind the use of EMA in place of storing logits.
>
> >The submission mentions that code will be released, but it is not included for review.
>
> We plan to release the code upon acceptance.  We are happy to release earlier in case it is necessary. Please let us know.
>
> >It might be best to place all tables/figures at the top or bottom of the page..
>
> Thank you for the suggestion. We will update the formatting in the final version.
>
> >"one such knowledge bases is"
>
> Fixed.
>
> We once again thank the reviewer for detailed feedback. We have made an utmost effort to address all the concerns raised. Please let us know in case we have missed something.

---

> > ### Comment · Reviewer_HdW5 · 2022-11-16
> > **Thank you for your thorough reply**
> >
> > I have read the authors' response in detail, as well as the revisions in their manuscript, which the authors kindly highlighted in blue. I re-state below my three main concerns and discuss how the authors have addressed them, along with my updated thoughts.
> >
> > 1. Placement with respect to the literature
> >    1. Related work discussion: the authors added multiple new references and adequately described their relations to the proposed approach. This is, to me, the most significant change as compared to the original manuscript.
> >    2. Baseline selection: the authors added three new baselines to the experiment in Figure 3---PackNet (a fixed-capacity approach), PAE (an expansive approach), and CPG (another expansive approach). As expected, the latter two were closest in performance to TAMiL, with CPG and TAMiL alternating for top performance throughout the sequence of tasks, finally converging to nearly identical performance after 20 tasks. However, no part of the new results is discussed in the manuscript (it simply states that TAMiL outperforms fixed-capacity models, but nothing about how it fares against dynamic-capacity models). There's also no comparison in the number of parameters of TAMiL with respect to PAEs or CPG, nor a description of how the main hyperparameters were chosen for them (e.g., the "target accuracy" for CPG). These discussions are critically important to assess the value of this empirical comparison.
> > 2. Use of task-specific parameters in baselines: the authors added an experiment in Table 6 to study the difference in using TAMs vs task-specific output heads in three of the replay-based baselines. While the authors claim that this resulted in "discernible performance improvement", it is unclear how significant these results are. How many random seeds were used for the experiment, what is the $\pm$ representing? It seems that only one of the three improvements is statistically significant, according to the authors' own reported errors. It's also unclear what the difference in the number of parameters is: are TAM-based methods using many more parameters than multi-head methods? Would it be possible to make them comparable, e.g. by adding more than just one task-specific layer for multi-headed methods? Also, it's not specified what the experimental setting is for these results; which data set was used?
> > 3. Clarity of the task inference method: the authors provided some clarification (in their response, not the revised manuscript) of how task inference was performed. My understanding of this method is that, during training of each task $t$, the method replays data from tasks $m<t$ via standard rehearsal. However, instead of using the correct task label $m$, the agent finds the $m$ such that the modulated features (those output by the corresponding TAM) most closely match the original features. This seems to indicate that, for each data point, we should expect the correct TAM to modulate the "irrelevant" features and pass along only the relevant ones (with irrelevant features presumably being those with low values). Is this correct so far? Now, what I still don't understand, and I don't see anything in the manuscript or the authors' response to address this, is how this encourages appropriate selection of the TAM. If the incorrect TAM is chosen during training, we might imagine that the predicted class will likely be incorrect, and the error will backpropagate through both the wrong task's TAM and the shared feature representation. How does this eventually lead to the correct TAM being chosen? And how does it avoid forgetting if the wrong TAM is chosen during training, since replay would be done incorrectly? I'm not saying the method is not good, I just would like to see a clearer description of why it should work.
> >
> > Overall, I found the authors' response useful for discussion, but still a bit far from turning the submission into a clear accept. That being said, given the improved placement with respect to the literature, especially in the discussion in Section 2, I am now slightly leaning towards acceptance and will update my score accordingly after the discussion period.

---

> > > ### Author Response · Authors · 2022-11-17
> > > **Reply to the reviewer HdW5**
> > >
> > > We thank the reviewer for taking the time to provide detailed feedback on our response. Our responses to the questions are as follows:
> > >
> > > >1. Placement with respect to the literature
> > >
> > > - Thank you for acknowledging the changes to the related works and consequent improved placement with respect to the literature.
> > >
> > > - Baseline selection:
> > > Thank you for pointing that out.  We have updated Figure 2 and associated text with PackNet, PAE and CPG.  We regret the lack of clarity with regard to chosen hyperparameters and growth in the number of parameters. We update the manuscript to reflect these suggestions. As we replicate the results reported in CPG, we omit hyperparameter selection and redirect the readers towards Table 1 in CPG for further information.
> > >
> > > >2. Use of task-specific parameters in baselines
> > >
> > > We employed three rehearsal-based approaches to showcase how TAMs perform in comparison to task-specific parameters in terms of multi-head. The results are an average of three runs and text after ‘±’ represents the standard deviation.  As rightly pointed out by you, TAMs and multi-head perform comparably with TAMs producing a slightly higher performance. Therefore, we update the manuscript and remove reference to "discernible performance improvement". We have updated the experimental settings as well. We also add another multi-head setting with a number of parameters in task-specific head comparable with that of TAMs. Even with such a setting, Task-IL performance is still comparable with TAMs.
> > >
> > > It is important to note that multiple-heads in Task-IL bring limited performance improvement compared to single-head. However, multiple-heads require task identity at inference time and do not work in Class-IL scenario. This is also true for several progressive networks (PNN, CPG, PAE) considered in this work. This limits their applicability to real life scenarios. On the other hand, TAMiL performs extremely well in both Class-IL and Task-IL without having to change anything in the proposed method. Therefore, we argue that TAMs bring more sophistication and design flexibility than task-specific counterparts in other approaches.
> > >
> > > >3. Clarity of the task inference method
> > >
> > > Yes, your understanding of the proposed method is correct thus far. We have updated Appendix E.5 with intuition behind working of the ignition event in TAMiL. As you have pointed out, selection of wrong TAM normally leads to wrong prediction and high loss. The ignition event in the proposed method relies completely on the supervisory signal from cross-entropy loss and consistency regularization. Therefore, both selected TAM and the shared backbone will be penalized for the wrong selection during backpropagation.   We agree that there is no explicit incentive for the proposed ignition event to select the right TAM. As selecting an appropriate TAM ensures a great reduction in loss for the buffered samples, we believe that over the course of training, CL model will be encouraged to select the right TAM to reduce the training loss, albeit with exceptions.  We acknowledge that explicit incentive might be necessary to further improve the proposed ignition event.
> > > We agree that proposed ignition event runs a risk of wrong TAM being selected and overwritten, and indeed results in forgetting of information pertaining to older tasks. It is also evident in substantial performance difference between TAMiL (Oracle) and TAMiL. TAMiL (Oracle) was trained completely devoid of ignition event and employs task identity to identify the right TAM during training and inference. As there was no mixing of information during training, TAMiL (Oracle) overcomes catastrophic forgetting to a great extent. As future work, we plan to focus on developing more accurate ignition events to match the performance of TAMiL (Oracle).
> > >
> > > We once again thank the reviewer for detailed feedback on our manuscript.

---

> ### Author Response · Authors · 2022-11-11
> **Reply to the reviewer HdW5**
>
>
> >Some details about the experimental evaluation are missing, which makes me wonder about the fairness of the comparisons
>
> We thank the reviewer for pointing that out. We have updated Appendix E.2 to include more information about the training and evaluation regime. We would like to clarify that we use the same evaluation settings (Class-/Task-IL, datasets, backbone, number of epochs, single linear classifier) as state-of-the-art methods considered in this work. Therefore, the results mentioned in this paper are directly comparable to those reported in their original manuscripts.
> >I wasn't able to follow exactly how the approach automatically infers the task ID during evaluation, and I believe that additional explanation would be needed ..
>
> The TAMs in our framework act as a communication bottleneck and select features relevant for the corresponding task. However, association between an input sample and its corresponding TAM is not given as task identity is not available during inference in Class-IL.  Motivated by ignition event in the brain (Appendix A.1), we develop a simple ignition event to select appropriate TAM both during training and inference. To this end, during training, each TAM first learns task-specific attention using task-specific data $D_t$. As our method employs experience-rehearsal, we use $D_m$ to automatically select the appropriate TAM. Since each TAM is associated with a specific task, inferring a wrong TAM for the buffered samples can result in sub-par performance and higher penalty in terms of cross-entropy loss and consistency regularization. This way, the CL model is trained to first capture task-specific information and also learn the routing through buffered samples using an ignition event.
>
> CL models without TAMs (DER++, CLS-ER), already accumulate information in their common representation space that is sufficient for decent classification performance. TAMs, on the other hand, denoisify these features resulting in higher performance due to lessened interference.  We empirically found that deviating too much from common representation space features incurred higher interference and consequent forgetting in presence of TAMs. Therefore, the task-specific attention should be such that it promotes denoising, but not at the expense of features important for the current task. To this end, we proposed a simple matching criterion that dynamically selects a TAM that is most similar to common representation space features. For buffered samples, appropriate TAM is dynamically selected using Equation 5. Only the output of selected TAM is forward propagated to global workspace. We then compute cross entropy loss and consistency regularization, and backpropagate the errors.
>
> The obvious downside of such an approximation is a drop in performance.  Compared to Oracle version, TAMiL with ignition event described in Equation 5 produces a subpar performance. We note this obvious limitation in Appendix B. TAMiL can benefit from more accurate matching criterion to match the performance of Oracle. More complex alternatives such as learning a policy using reinforcement learning, a gating mechanism using Gumbel-softmax, and prototype matching can also be explored in place of proposed matching criterion to further improve the selection accuracy.
>
> As suggested by the reviewer, there are several techniques for automatically determining the task at inference time. As a future work, we envisage to look into ways to effectively emulate Oracle version to bridge the gap between Class-IL and Task-IL in CL.
>
> >Would the authors agree with that statement?
>
> We broadly agree with the reviewer that our core contributions are threefold: Developing a CL method motivated by GWT that is robust, well calibrated, and scalable without requiring task-identity at inference time. To this end, we propose task-specific attention analogous to communication bottleneck in the brain. We respectfully disagree that TAMs are mere task-specific parameters. TAMs act as feature selectors and play a key role in mitigating task interference.  Appropriate choice of TAMs is important for ensuring a good performance in CL (Table A.2). Finally, we also propose a simple matching criterion to dynamically select appropriate TAM during inference time.
>
> We qualitatively and quantitatively compare with some of the approaches suggested by the reviewer. Accordingly, we expanded our related works and experiments in the updated manuscript.
>
> Furthermore, TAMs are a plug-and-play component of TAMiL which can be used with ease to benefit any rehearsal-based approaches (Figure 3 (left)). In entirety, TAMiL brings forth a novel approach for mitigating catastrophic forgetting in CL through task-attention.  As pointed out by the reviewer, this is an alternative to multiple classification heads that warrants further investigation in the literature.

---

> ### Author Response · Authors · 2022-11-11
> **Reply to the reviewer HdW5**
>
> Summary  :
>
>  We proposed TAMiL, , a CL method that, while drawing inspiration from GWT, successfully combines rehearsal and parameter isolation with little memory overhead (Table 4), superior performance (Table 1) without capacity saturation (Figure 2), and without requiring task-identity at inference time. Our main motivation was to develop a task-specific attention mechanism that leverages experience-rehearsal and is supported through maximum re-usability and scalable neurogenesis. To this end, we compare with the state-of-the-art rehearsal-based methods and parameter-isolation (evolving architectures) methods. The relevance of each of these related works is as follows: TAMiL employs rehearsal with reservoir sampling with a single expanding head representing all classes in both Class-IL and Task-IL settings. Therefore, we compare TAMiL with several recent rehearsal-based methods.  Since TAMiL involves task-specific attention modules, we consider parameter isolation methods for comparison. Among parameter isolation, dynamic sparse networks work within a fixed model capacity while progressive networks expand model capacity either naively or intelligently using some criterion.
>
> Although TAMiL grows in size, the growth in number of parameters is negligible compared to the size of the base CL model (Table 4). Secondly, TAMiL has a plug and play component which can be used with ease to benefit any rehearsal-based approaches (Figure 3 (left)). Thirdly, TAMiL does not require task-identity at inference time. Although, its availability would further improve TAMiL (Oracle version), it is not a must to ensure good performance. Finally, TAMs in TAMiL aren’t any additional parameters that grow / shrink based on task-identity. TAMs are simple, fixed size, compact feature selectors that inhibit or activate important features to reduce catastrophic forgetting in CL. As pointed out by the reviewer, we have conducted extensive empirical evaluation to provide the readers with detailed insights on the working of TAMiL.
>
> >There are a number of references that should likely be included and compared against qualitatively (at least) or quantitatively (ideally)
>
> We thank the reviewer for providing references to broader related works in evolving architecture literature. We agree with the reviewer that many of these works address some of the issues TAMiL is trying address and have an overlap in their approach in doing so. Specifically, some of these approaches are more efficient than PNNs or grow drastically slower while scalable. As per reviewer’s suggestion, we update our manuscript to discuss some of these approaches in the ‘evolving architectures’. We have updated Figure 2 by adding comparison with more related works.

---

### Official Review · Reviewer_yHwh · 2022-10-26

**Confidence:** 4
**Correctness:** 4
**Technical Novelty And Significance:** 3
**Empirical Novelty And Significance:** 3
**Recommendation:** 6

**Clarity, Quality, Novelty And Reproducibility:**

Clarity:
The paper needs more clarity in terms of usage in practical life.

Quality:
The quality of content and work is good.

Reproducibility:
Experiment details are given in a way to compelement publicly available code, data later (if accepted).

Novelty:
Among the 4 contributions listed, some of them can be combined into 2 - or provide more support.

**Details Of Ethics Concerns:**

Docoloc plagiarism check is 11 % - which is at borderline, but acceptable given the quality of content.

**Strength And Weaknesses:**

trengths:
1. The paper is written well
2. The approach seems to be novel, though needing some polishing in description
3. Experimental results are supportive of claims
4. Code / data to be released later
5. Model calibration section
6. Table 1, Figure 8

Weakness:
1. The contributions of the paper need more clarity with examples
2. The gaps wrt prior art in Related work section should be highlighted instead of merely writing a line
3. The ignition event (eqn 5) matching criteria needs some defense in light of Juliani et al., 2022.
4. The task action space can be elaborated - it is not clear
5. List down the limitations and assumptions clearly (like Given a sufficiently high buffer size, our method outperforms PNNs)

**Summary Of The Paper:**

Inspired by Global Workspace Theory of conscioussness authors propose TAMiL - a continual learning method that entails task-attention modules to capture task-specific information from the common representation space. Experimental results show that their method outperforms SOTA rehearsal-based and dynamic sparse approaches, while being scalable. The paper also claim to  mitigate catastrophic forgetting along with reducing task-recency bias.

**Summary Of The Review:**

The paper after polishing can be considered for ICLR.

Suggestions:
1. Abstract should be to the point and supported by a line of results
2. The related work section should follow the current trend - topic-wise listing of prior art
3. Round down to a specific problem you are trying to solve and then think about generalization
4. List down the assumptions - ex. the task is highly dependent on perception module
5. GWT is quite old, could have also looked at Michael Graziano's attention schema
6. For GWT, can look at references of this work: https://www.frontiersin.org/articles/10.3389/fpsyg.2021.749868/full
7. Fig 6 supplementary, the feature extractors and selectors can be explained along with choice of cosine sim
8. : The early layers of DNNs capture generic information, while the later layers capture task-specific information. - need details
9. Our CL model consists of as many TAMs as the number of tasks. - any better way?

Miscellaneous:
1. Optional - the title looks odd due to hypen break
2. Break long sentences in smaller forms
3. The prevalence of several knowledge bases in the brain - correct?
4. Too much - Inspired by GWT

---

> ### Author Response · Authors · 2022-11-09
> **Reply to the Reviewer yHwh**
>
> We thank the reviewer for taking the time to review our work and providing detailed feedback. Our responses to the questions are as follows:
>
> > The contributions of the paper need more clarity with examples
>
> We have updated the contributions with example scenarios and prior art. Please let us know in case it needs further clarification.
>
> >The gaps wrt prior art in the Related works section should be highlighted instead of merely writing a line. The related work section should follow the current trend - topic-wise listing of prior art
>
> Based on the suggestions of all reviewers, we have updated the related works to reflect closely related prior arts. In addition, we segregate the related works into two sections namely rehearsal-based and evolving architectures. We also update the limitations in each of these sections to highlight the gap TAMiL is trying to address.
>
> > The ignition event (eqn 5) matching criteria needs some defense in light of Juliani et al., 2022.
>
> We have updated the manuscript to include more intuition behind ignition event in TAMiL in Appendix A.1. Combining inputs from multiple reviewers, we elaborate on how ignition event plays a vital role in the activation and inhibition of a subset of workspace neurons both in the brain and TAMiL. The ignition event here is mainly concerned with TAMs and their activation and inhibition based on the task identity of the incoming sample. We selected the matching criterion in Equation 5 as an ignition event due to its simplicity and lack of additional trainable parameters. However, complex alternatives such as learning a policy using reinforcement learning, a gating mechanism using Gumbel-softmax, and prototype matching can also be explored.
>
> >The task action space can be elaborated - it is not clear
>
> We are not sure whether we understood this correctly. We have updated section 3.4 to add more clarity on how TAMs attend to common representation space features. Please let us know more information on this if we haven't yet addressed it properly.
>
> > List down the limitations and assumptions clearly (like Given a sufficiently high buffer size, our method outperforms PNNs). List down the assumptions - ex. the task is highly dependent on the perception module
>
> Based on the reviewer's feedback, we update section 3.3 with assumptions and Appendix B with limitations. However, there was a miscommunication on our part in the above claim "Given a sufficiently high buffer size, our method outperforms PNNs". TAMiL employs a memory buffer of size 500 and reports Task-IL accuracy slightly lagging behind PNNs. We intend to report that given a  buffer size slightly higher than 500, TAMiL can outperform PNNs (clear from Seq-CIFAR100 Task-IL results). Compared to the dataset sizes, buffer size 500 is relatively small. We regret the choice of words and update the manuscript to reflect this understanding.
>
> > Abstract should be to the point and supported by a line of results
>
> In our current version of the abstract, we highlight that TAMiL "outperforms state-of-the-art rehearsal-based and dynamic sparse approaches  while being well-calibrated with reduced task-recency bias." Please let us know if you have any suggestions regarding how we can bring more clarity to our abstract.
>
> > Round down to a specific problem you are trying to solve and then think about generalization
>
> Aligning with your previous comments, we have sharpened our related works to highlight the problems in the prior art. In the final paragraph of the related works in the updated manuscript, we highlight how TAMiL is placed in the literature to address these shortcomings. Please let us know if any more information is necessary to highlight the problems in the prior art.
>
> >GWT is quite old, could have also looked at Michael Graziano's attention schema
>
> Attention schema theory (AST) [4, 6] proposed by Michael Graziano posits that the brain controls its own attention partly by constructing a model of attention. The proposed “attention schema,” analogous to the body schema, is a constantly updating set of information that represents the basic functional properties of attention, represents its current state, and makes predictions such as how attention is likely to transition from state to state or to affect other cognitive processes.
> AST disentangles how awareness and attention are related. As the brain may have evolved a model of attention, developing such an attention schema in DNNs is not straightforward.  On the other hand, GWT provides a simpler model of conscious information access in the brain with discernible implications for continual learning. Therefore, we leverage GWT as inspiration for developing TAMiL, a sophisticated brain-inspired continual learner.
> As there is increasing validation for AST in the literature, we agree with the reviewer that the AST-inspired system might unlock robust attention traits in artificial continual learners.

---

> > ### Author Response · Authors · 2022-11-09
> > **Reply to the Reviewer yHwh**
> >
> >
> > > For GWT, can look at references of this work: https://www.frontiersin.org/articles/10.3389/fpsyg.2021.749868/full
> >
> > We thank the reviewer for the relevant reference for GWT. This is another paper from Baars on the latest developments in GWT. We have updated our references accordingly.
> >
> > > Fig 6 supplementary, the feature extractors and selectors can be explained along with a choice of cosine sim
> >
> > We have updated Appendix A.3 to provide information about feature selectors, extractors, and cosine similarity.
> >
> > > The early layers of DNNs capture generic information, while the later layers capture task-specific information. - need details
> >
> > We agree with the reviewer that more information might be necessary to justify this claim in the paper.  Several works [2, 3] have found that memorization normally happens in the later layers due to decreasing manifold dimension and radius, whereas earlier layers ignore spurious noise in the data and instead learn generalizable features. We bank on this analytical analysis and hypothesize that redundancy might be necessary for the later layers to reduce interference.  We have updated the manuscript to reflect this understanding.
> >
> > > Our CL model consists of as many TAMs as the number of tasks. - any better way?
> >
> > Motivated by GWT, we set out to present a novel approach to mitigate catastrophic forgetting by attending to task-specific features from the common representation space. We show that such an approach outperforms several prior rehearsal-based, parameter-isolation, and dynamic sparse networks. Although the number of TAMs grows commensurate with the number of tasks, it has very little impact in terms of the number of parameters (refer to Table 4 in the updated manuscript).
> > We agree with the reviewer that this can be further optimized by employing a single TAM for multiple tasks. However, one needs to be cautious about interference between tasks that are merged into a single TAM. In future work, we intend to leverage task similarity to merge multiple TAMs into one without losing performance.
> >
> > > Optional - the title looks odd due to hyphen break
> >
> > We agree with the reviewer that the hyphen break in the title makes it a bit odd. We were unable to fix it as it concerns ICLR styling. Please let us know if you have any suggestions in mind.
> >
> > > Break long sentences in smaller forms
> >
> > Thank you for pointing that out. We believe it arises from our writing style. We will keep this in mind and try to ensure we stick to smaller sentences in the final revision.
> >
> > > The prevalence of several knowledge bases in the brain - correct?
> >
> > Kudithipudi et al (2022) [1] states that “self-regulated neurogenesis plays a role in scaling up the number of new memories that can be encoded and stored during one’s lifetime without catastrophic forgetting of previously consolidated memories.”. We refer to these multiple memories as knowledge bases. We agree with the reviewer that this can be misleading. Therefore, we omit this reference pointed out by the reviewer.
> >
> > > Too much - Inspired by GWT
> >
> > We are not sure whether we understood the suggestion correctly. Are you referring to the repetition of the word 'Inspired by GWT' or lot of references to GWT as a motivation for our work? Please let us know.
> > Our manuscript describes GWT in detail to provide a more comprehensive understanding of its background and its relevance for CL. However, we agree that some of the information about GWT can be re-directed toward related literature with appropriate citations. Therefore, we have reduced some of the texts pertaining to GWT in the method section and moved them to Appendix C.
> >
> > We once again thank the reviewer for detailed and insightful feedback. Please let us know in case we have missed any open points.
> >
> > [1] Kudithipudi, Dhireesha, et al. "Biological underpinnings for lifelong learning machines." Nature Machine Intelligence 4.3 (2022): 196-210.
> >
> > [2] Baldock, Robert, Hartmut Maennel, and Behnam Neyshabur. "Deep learning through the lens of example difficulty." Advances in Neural Information Processing Systems 34 (2021): 10876-10889.
> >
> > [3] Stephenson, Cory, et al. "On the geometry of generalization and memorization in deep neural networks." International Conference on Learning Representations. 2020.
> >
> > [4] Graziano, Michael SA, and Taylor W. Webb. "The attention schema theory: a mechanistic account of subjective awareness." Frontiers in psychology (2015): 500.
> >
> > [5] Wilterson, Andrew I., and Michael SA Graziano. "The attention schema theory in a neural network agent: Controlling visuospatial attention using a descriptive model of attention." Proceedings of the National Academy of Sciences 118.33 (2021): e2102421118.
> >
> > [6] Graziano, Michael SA. Consciousness and the social brain. Oxford University Press, 2013.

---

> ### Author Response · Authors · 2022-11-17
> **General reply to the reviewer yHwh**
>
> As the discussion stage-1 is coming to an end in a day, kindly let us know whether we have addressed all your concerns. We would be happy to discuss more if there are any open questions. Thank you in advance.

---

### Official Review · Reviewer_JQ6f · 2022-10-29

**Confidence:** 2
**Correctness:** 3
**Technical Novelty And Significance:** 3
**Empirical Novelty And Significance:** 3
**Recommendation:** 8

**Clarity, Quality, Novelty And Reproducibility:**

The work is mostly clearly communicated, though it would be even better if Figure 1 could be referred to more frequently in Section 3 of the main text. For example, the color coding in Figure 1 wasn’t very clear to me and I couldn’t find much detail about it in the main text.

The work combines two common approaches in continual learning, namely replay and regularization, thus is quite novel.

The training details are provided in the appendix, thus the work should be reproducible upon code release.

**Strength And Weaknesses:**

Strengths:
- This paper is well-written and the figures are easily digestible. The baseline models included a wide range of selections, and varied in buffer sizes. TAMiL applies the global workspace theory, a longstanding neuroscience theory for consciousness, to the continual learning setting, which is quite a novel approach.

Weaknesses:
- The concept of global workspace is influential to the field of cognitive neuroscience, and this paper shows great novelty by taking inspiration from it. However, exactly how the global workspace is mathematically defined, constructed and used was not explained well enough in this paper, unlike the common representation space which the author explains in great detail. Moreover, since the global workspace theory has been linked to many neuroscience findings [(Mashour et al., 2020)](https://www.sciencedirect.com/science/article/pii/S0896627320300520), it would be interesting to draw potential connections between TAMiL and the neural circuits underlying the ignition event.

Questions:
- Figure 1 bottom: is $L_p$ the same as $L_{pd}$, i.e. the pairwise discrepancy loss?
- What are the transformation coefficients mentioned in section 3.4 second paragraph, and where does it fit in Figure 1?


**Summary Of The Paper:**

In this paper, the authors introduced TAMiL, a continual-learning model inspired by the global workspace theory that can learn multiple tasks without catastrophic forgetting by constructing a common representation space across tasks. By combining previous approaches on self-regulated neurogenesis and experience replay, TAMiL outperformed current state-of-the-art rehearsal-based methods as well as popular regularization-based methods on Seq-CIFAR10, Seq-CIFAR 100 and Seq-TinyImageNet, both in Class-Incremental Learning setting and and Task-incremental Learning setting. The basic unit of TAMiL, TAM, can also be flexibly augmented to previous rehearsal-based methods to boost performance.

**Summary Of The Review:**

I lean slightly towards accepting this paper for ICLR: the proposed model, inspired by the global workspace theory, robustly outperforms state-of-the-art continual learning models in many settings. Ablation experiments also provided insights into the importance of each of the components of the model.

---

> ### Author Response · Authors · 2022-11-09
> **Reply to the Reviewer JQ6f**
>
> We thank the reviewer for the constructive comments and especially the recent reference on the global neuronal workspace hypothesis. Our responses to the questions are as follows:
>
> Global workspace theory (GWT) of conscious information processing in the brain was first proposed by Baars (1988) [1].  Inspired by GWT of conscious information access in the brain, we proposed TAMiL to mitigate catastrophic forgetting in DNNs in CL. TAMiL distinguishes the encoder output into a common representation space and global workspace by inserting task-attention modules, the core component of TAMiL. As pointed out by the reviewer, we have added additional clarification in Section 3.1 to clarify the construct behind global workspace.
>
> GWT is a psychological construct arguing that only the behaviorally relevant information from the perceptual contents in the common representation space is admitted to the global workspace when warranted by a task. On the other hand, the Global neuronal workspace (GNW) hypothesis [2] posits that the brain entails a second computational space composed of widely distributed excitatory neurons that selectively mobilize or suppress, through descending connections, the contribution of specific processor neurons. GNW is associated with an ignition event that is characterized by sudden, coherent, and exclusive activation and inhibition of workspace neurons [3]. Analogously, TAMiL also entails an ignition event wherein an appropriate task-attention module (TAM) is activated while the rest of TAMs are inhibited when warranted by a specific task. We agree with the reviewer that parallels can be drawn between ignition events in TAMiL and GNW. We have updated Appendix A1 to provide more intuition about the ignition event in TAMiL.
>
> >The work is mostly clearly communicated, though it would be even better if Figure 1 could be referred to more frequently in Section 3 of the main text. For example, the color coding in Figure 1 wasn’t very clear to me and I couldn’t find much detail about it in the main text.
>
> Thank you for pointing that out. We have updated Figure 1 with clearer explanations and color coding for better understanding of the proposed method. Also, we added more references to Figure 1 in more places in Section 3.
>
> > Figure 1 bottom: is $L_p$ the same as $L_{pd}$, i.e. the pairwise discrepancy loss?
>
>  Yes, both denote the pairwise discrepancy loss. We have updated Figure 1 accordingly.
>
> >What are the transformation coefficients mentioned in section 3.4 second paragraph, and where does it fit in Figure 1?
>
> We refer to the output of task-specific TAMs as transformation coefficients. The output of any TAM learns to select/attend to features important for the corresponding task. Specifically, they assign a weight between 0 and 1 to each incoming feature. We refer to these attention vectors as transformation coefficients. These coefficients are then applied to the features of the common representation space using element-wise multiplication. We have updated Section 3 to reflect the same.
>
> We once again thank the reviewer for insightful feedback. We have made the utmost effort to address all the concerns raised. Please let us know in case we have missed any open points.
>
> [1] Baars, Bernard J. "A cognitive theory of consciousness." (1988).
>
> [2] Dehaene, Stanislas, Michel Kerszberg, and Jean-Pierre Changeux. "A neuronal model of a global workspace in effortful cognitive tasks." Proceedings of the national Academy of Sciences 95.24 (1998): 14529-14534.
>
> [3] Dehaene, Stanislas, Claire Sergent, and Jean-Pierre Changeux. "A neuronal network model linking subjective reports and objective physiological data during conscious perception." Proceedings of the National Academy of Sciences 100.14 (2003): 8520-8525.

---

> > ### Comment · Reviewer_JQ6f · 2022-11-30
> > **Thank you authors for your response**
> >
> > I appreciate the clear explanation on the distinction between GWT and GNW and how TAMiL is related to these concepts. I find the addition of Appendix A1 and the additional explanations in Section 3.1 very useful for me to understand where TAMiL stand in relation to the neuroscience and psychology literature. The authors also addressed my other questions and comments clearly. Overall I think the revised manuscript is now stronger than original, and should be accepted for ICLR. I will update my score accordingly.

---

> ### Author Response · Authors · 2022-11-17
> **General reply to the reviewer  JQ6f**
>
> As the discussion stage-1 is coming to an end in a day, kindly let us know whether we have addressed all your concerns. We would be happy to discuss more if there are any open questions. Thank you in advance.

---

### Author Response · Authors · 2022-11-11
**Reply to all reviewers**

Based on all reviewers' feedback, the following modifications have been made to the manuscript:

- Introduction
  * Added clarifications regarding attention in transformers,
  * Updated the contributions with examples
- Related Works
  * Clarified the major drawbacks in the existing rehearsal-based approaches
  * Added more methods under 'Evolving architectures'
  * Clarified how TAMiL addresses gaps in the related works

- Proposed Method
  * Updated the caption for the Figure 1 and referenced more in this section
  * Mathematically defined the construct behind global workspace
  * Added assumptions and clarification for auto-encoder
  * Clarified the intuition behind placing TAM higher up in the layer hierarchy.
  * Clarified transformation coefficients

- Results
  * Updated Figure 2 with more baselines
  * Added more information about task probabilities

- Appendix: Due to space Issues, we have added more information requested by reviewers in the Appendix.
  * A.1  - Ignition event analogy from Global neuronal workspace theory
  * A.3  - TAMs similarity
  * B  - Limitations
  * C  - Prominence of GWT in CL
  * D.4  - Comparison with multi-head setup
  * E.2  - Clarifications regarding experimental setup
  * E.3  - backbones used for comparison with dynamic sparse networks
  * E.4  - Relevance of EMA and how we use it in TAMiL

Please let us know in case we have missed any of your feedback.

---

### Author Response · Authors · 2022-11-16
**General reply all reviewers**

As the deadline for discussion stage-1 is fast approaching, Kindly let us know whether we have addressed all your concerns. We would be happy to discuss more if there are any open questions. Thank you in advance.

---

### Decision · Program_Chairs · 2023-01-20

**Decision:**

Accept: poster

**Justification For Why Not Higher Score:**

Despite feeling overall positive about this paper, I think it still has some weaknesses that prevent me from recommending spotlight or oral.

**Justification For Why Not Lower Score:**

I believe this paper is of interest to the research community and presents an interesting angle between continual learning and GWT, so I'd like to see it accepted.

**Metareview: Summary, Strengths And Weaknesses:**

This paper proposes a continual learning model (TAMiL) inspired by the global workspace theory that can learn multiple tasks without catastrophic forgetting by constructing a common representation space across tasks. By combining previous approaches on self-regulated neurogenesis and experience replay, TAMiL outperforms current state-of-the-art rehearsal based methods and regularization based methods on existing benchmarks. This is a borderline paper that triggered some questions from the reviewers, regarding the need for clarifying some of the connections to the neuroscience literature and connection to the literature on continual learning. The authors addressed most of the concerns of the reviewers in their amswers and in the revised version. I recommend acceptance.


**Note From Pc:**

if the above contains the word "oral" or "spotlight" please see: "oral" presentation means -> notable-top-5% and "spotlight" means -> notable-top-25%. As stated in our emails, we are disassociating presentation type from AC recommendations

**Summary Of Ac-Reviewer Meeting:**

Unfortunately reviewers were not responsive to my attempt to trigger discussion and to schedule a meeting (with the exception of one reviewer, who engaged in discussion with the authors and was ready to meet). In the discussion phase some reviewers increased their scores and update their reviews. This makes me lean towards acceptance.